# CHART DEEP RESEARCH IN LVLMS VIA PARALLEL RELATIVE POLICY OPTIMIZATION

**Jiajin Tang[1,2]**∗, **Gaoyang[1]**†, **Wenjie Wang[2]**, **Sibei Yang[3]**‡, **Xing Chen[1]**,
[1]ByteDance,  [2]ShanghaiTech University,
[3]School of Computer Science and Engineering, Sun Yat-sen University

## ABSTRACT

With the rapid advancement of data science, charts have evolved from simple numerical presentation tools to essential instruments for insight discovery and decision-making support. However, current chart data intelligence exhibits significant limitations in deep research capabilities, with existing methods predominantly addressing shallow tasks such as visual recognition or factual question-answering, rather than the complex reasoning and high-level data analysis that deep research requires. This limitation stems from two primary technical bottlenecks: at the training level, existing post-training techniques exhibit deficiencies in handling multi-dimensional reward signal interference and heterogeneous data gradient conflicts, preventing models from achieving balanced development across multiple capability dimensions; at the evaluation level, current methods remain limited to factual retrieval and basic computation, failing to assess end-to-end analytic reasoning and other deep research capabilities. To address the training challenge, we propose PRPO, which performs parallel optimization across reward dimensions and capability partitioning across data types, effectively disentangling conflicts between heterogeneous data and multi-dimensional reward signals while ensuring optimization stability. For the evaluation challenge, we construct MCDR-Bench based on the "error uniqueness principle," transforming subjective generation assessment into objective error identification through controllable error injection, enabling quantifiable evaluation of deep research capabilities. Experimental validation confirms that the proposed PRPO and MCDR-Bench jointly establish a unified framework that systematically advances chart deep research through enhanced collaborative training and objective evaluation.

## 1 INTRODUCTION

Data-driven technologies and multimodal large models have elevated charts from simple "numerical presentations" to essential tools for discovery and decision-making (Kim et al., 2020; Masry et al., 2024; Hoque et al., 2022). The progress of multimodal large language models (MLLMs) (Bai et al., 2025; Achiam et al., 2023; Comanici et al., 2025; Anthropic, 2025b; Grattafiori et al., 2024) has transformed charts into core analytical instruments for pattern discovery, hypothesis testing, and strategic decision support across finance, healthcare, business, and scientific domains (Lee et al., 2025; Shwartz-Ziv & Armon, 2022; Shigarov, 2023).

However, advancing chart deep research capabilities—the ability to conduct sophisticated analytical reasoning, pattern synthesis, and strategic insights from visual data—remains a fundamental challenge in current AI systems. While charts have evolved into sophisticated analytical instruments, existing methods predominantly address shallow tasks such as visual recognition or factual question-answering (Masry et al., 2022; Wang et al., 2024; Xu et al., 2023), falling short of the complex reasoning and high-level analytical processes that define genuine deep research capabilities (Masry et al., 2025; Lin et al., 2025; Mathew et al., 2022).

---

∗Work done during an internship at ByteDance.

†Project lead.

‡Corresponding author is Sibei Yang.

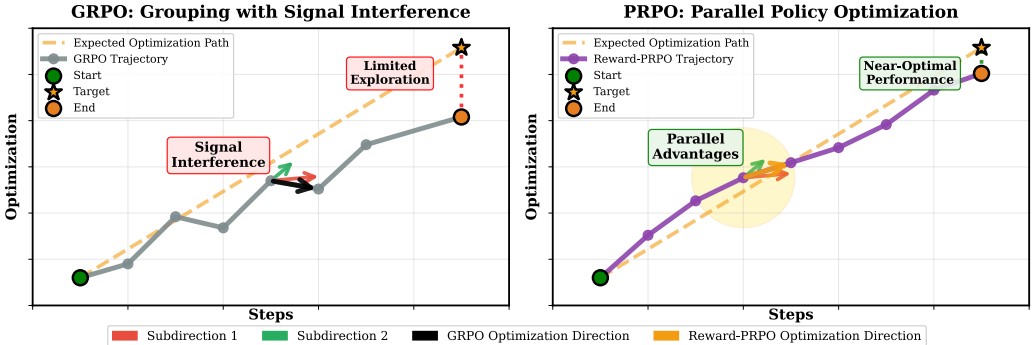

Figure 1: Optimization trajectories of GRPO vs. PRPO. Left: under multi-dimensional rewards, GRPO suffers from signal interference and limited exploration. Right: PRPO decomposes optimization across reward dimensions and data types, reducing interference and enabling specialized learning, which yields more effective exploration and near-optimal performance.

The development of chart deep research capabilities is systematically constrained by two fundamental bottlenecks. At the training level, existing post-training techniques exhibit deficiencies in handling the multi-dimensional reward conflicts and heterogeneous data challenges inherent in developing complex analytical reasoning abilities. At the evaluation level, current assessment frameworks remain misaligned with deep research demands, lacking methodologies to evaluate the end-to-end analytic reasoning that defines advanced chart analysis capabilities.

**Training Bottleneck for Deep Research Capabilities.** Developing robust chart deep research capabilities requires sophisticated training methodologies that can handle complex multi-dimensional learning. Chart deep research encompasses diverse analytical tasks such as background knowledge integration, fact extraction, relationship construction, deep report reasoning, and strategic forecasting—capabilities that demand coordinated development across multiple cognitive dimensions. However, as illustrated in Figure 1, existing preference-optimization methods such as Group Relative Policy Optimization (GRPO) (Shao et al., 2024) exhibit **conflict-constrained optimization dynamics** that hinder the development of these integrated capabilities. We observe two core limitations that specifically impede deep research capability development: (1) **Multi-dimensional reward interference**—aggregating heterogeneous signals leads to mutual cancellation of advantages and inhibits the exploration necessary for complex reasoning development; (2) **Data joint optimization conflicts**—gradients from different analytical dimensions may dominate or cancel each other, preventing the balanced capability development essential for deep research competency.

**Evaluation Bottleneck for Deep Research Capabilities.** Assessing chart deep research capabilities presents unique challenges that existing evaluation frameworks fail to address. Current benchmarks remain focused on surface-level tasks such as factual question answering or basic numerical operations (Masry et al., 2022; Wang et al., 2024; Xu et al., 2023; Masry et al., 2025; Lin et al., 2025; Mathew et al., 2022), failing to evaluate the end-to-end analytic reasoning, synthesis, and strategic insight generation that characterize advanced deep research capabilities. This evaluation gap—stemming from high annotation costs and subjective answer diversity—creates a critical barrier to systematically advancing and measuring progress in chart deep research.

To systematically advance chart deep research capabilities, we propose a unified framework addressing both training and evaluation bottlenecks. For training, we introduce **Parallel Relative Policy Optimization (PRPO)**, which enables the coordinated development of multi-dimensional analytical capabilities by performing parallel optimization across reward dimensions and capability partitioning across data types. For evaluation, we construct **MCDR-Bench** based on the *error uniqueness principle*, transforming subjective deep research assessment into objective error identification, enabling systematic measurement of analytical reasoning capabilities.

**Overall Contributions.** (1) **We systematically analyze the bottlenecks constraining chart deep research capability development**, identifying training challenges from multi-dimensional optimization conflicts and evaluation challenges from subjective assessment complexity. (2) We propose **PRPO**, enabling coordinated development of complex analytical capabilities through parallel optimization that mitigates reward interference in multi-dimensional training. (3) We present **MCDR-**

**Bench**, providing systematic evaluation of chart deep research capabilities through objective error identification methodology. (4) **Together, our unified framework establishes a systematic pathway for advancing chart deep research capabilities**, addressing both capability development and assessment challenges.

## 2 RELATED WORK

**Chart understanding benchmarks** have progressed from simple data extraction to advanced reasoning. Early benchmarks like PlotQA (Methani et al., 2020) and ChartQA (Masry et al., 2022) focused on factual question-answering and multi-step reasoning. Later work expanded to diverse visual representations through infographic integration (Mathew et al., 2022) and automatic chart summarization (Kantharaj et al., 2022; Tang et al., 2023). State-of-the-art benchmarks, including ChartBench (Xu et al., 2023), ChartQAPro (Masry et al., 2025), and InfoChartQA (Lin et al., 2025), improve evaluation quality by introducing diverse question types and reducing annotation bias. Nevertheless, most benchmarks remain limited to basic QA tasks, lacking evaluation of professional analyst capabilities such as deep insights, causal reasoning, and strategic decision-making—gaps that motivate our proposed deep research paradigm. Table understanding has followed a similar trajectory through multimodal frameworks (Zheng et al., 2024; Mathur et al., 2024; Titiya et al., 2025).

**Chart understanding algorithms** have evolved from visual alignment to specialized reasoning. Early work explored multimodal adaptation via domain-specific training (Han et al., 2023) and multi-task pattern recognition (Wei et al., 2024), while practical deployment drove efficiency optimization with techniques like visual token merging (Zhang et al., 2024a) and mixture-of-experts architectures (Xu et al., 2024). Recent methods target complex reasoning through human-like sketching (Huang et al., 2025) and structured chart representations for long-chain reasoning (Jia et al., 2025). Existing algorithms, however, largely focus on recognition and QA, leaving systematic modeling of professional analyst skills unaddressed.

**RLHF algorithms** have advanced from basic alignment to efficient optimization of language models using human feedback. PPO (Ouyang et al., 2022) stabilizes policy updates, while DPO (Rafailov et al., 2023) improves efficiency by avoiding explicit partition function computation. Later methods enhance training stability and sample efficiency: GRPO (Shao et al., 2024) removes independent value model training, entropy collapse is addressed by DAPO (Yu et al., 2025), replay mechanisms improve sample utilization (Li et al., 2025), and self-explanatory sample generation tackles low success rate scenarios (Zhou et al., 2025). Most recently, GSPO (Zheng et al., 2025) introduces sequence-level importance ratios with clipping and reward assignment, improving stability, efficiency, and performance for Mixture-of-Experts models.

While benchmarks and algorithms have become increasingly sophisticated, they still primarily target factual QA and basic chart recognition. Our paradigm innovates by systematically modeling professional analyst capabilities, enabling deep insights, causal reasoning, and strategic decision-making.

## 3 MCDR-BENCH

**Chart Collection**. Addressing the core problem of "high chart complexity but shallow understanding depth" identified in the Introduction, MCDR-Bench focuses on collecting complex charts capable of supporting deep research during the image collection phase. We carefully curate chart data from professional platforms including Dashboard Design Patterns (Bach et al., 2022), MME-realworld (Zhang et al., 2024b), ChartQAPRO (Masry et al., 2025) and Pew Research (Krogstad et al., 2016). To ensure data quality meets the requirements for deep research evaluation, we employ two key filtering criteria: first, we filter out overly simple charts (such as basic bar charts or pie charts with single data series) as these charts cannot demonstrate the true value of charts as "advanced analytical tools"; second, we prioritize complex charts containing multi-element, multi-layered information to ensure they possess the data richness necessary to support deep research tasks. We ultimately selected 1,021 high-quality charts that not only meet the complexity requirements of the visualization layer in form but also possess the intrinsic value to support deep understanding and analysis.

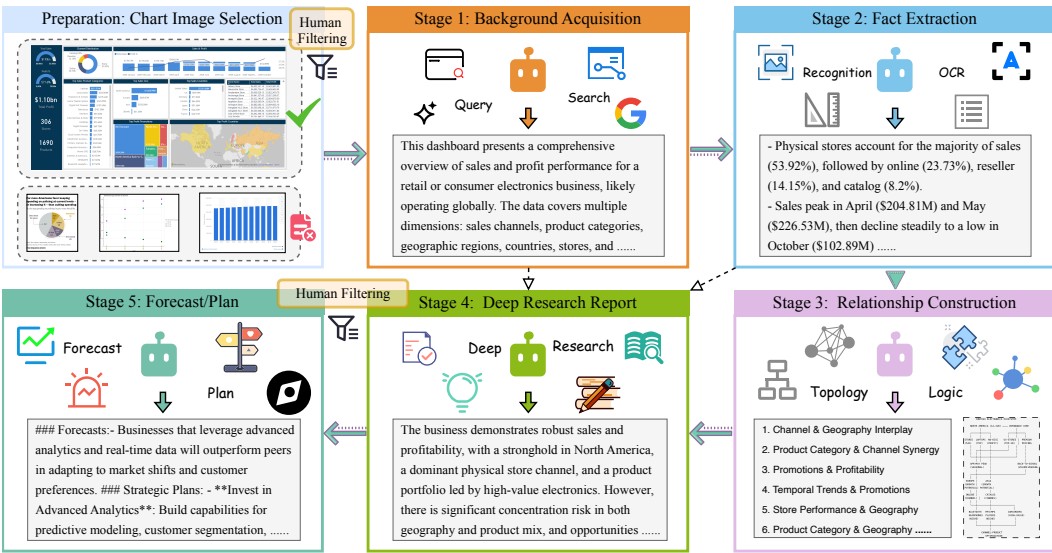

Figure 2: Multi-agent annotation process for multimodal chart deep research. The process consists of five stages: (1) **Background Acquisition**: retrieving domain-specific knowledge; (2) **Fact Extraction**: extracting atomic data elements; (3) **Relationship Construction**: modeling topological and logical connections; (4) **Deep Research Report Generation**: synthesizing comprehensive reports; (5) **Forecast/Plan**: proposing strategic recommendations. Human filtering ensures quality control throughout the process. The five specialized expert agents are represented by 🤖🤖🤖🤖🤖.

Our annotation framework employs a two-phase design to systematically evaluate chart analysis capabilities. **Phase 1 generates high-quality deep research reports**, while **Phase 2 converts subjective generation into objective error identification** through systematic error injection.

**Phase 1: Report Generation.** As shown in Figure 2, the generation process follows five interconnected stages: (i) Background Acquisition retrieves domain-specific context to enrich chart interpretation; (ii) Fact Extraction parses atomic data elements such as values and temporal sequences; (iii) Relationship Construction models intra- and inter-chart dependencies; (iv) Deep Research Report synthesizes trends, anomalies, and patterns into coherent analysis; and (v)Forecast/Plan derives forward-looking recommendations for decision support. Human review is specifically implemented at the critical report generation and decision-making stages, where human experts validate analytical accuracy and strategic soundness.

**Phase 2: Error Injection.** This phase transforms subjective report generation into objective error identification by introducing targeted perturbations corresponding to the five capabilities from Phase 1 (knowledge, facts, relationships, reporting, and forecasting). The process yields 3,084 high-difficulty samples with complex charts, enabling robust and fine-grained capability assessment. Complete construction details appear in Appendix E.

**Evaluation Paradigm.** MCDR-Bench delivers both a comprehensive dataset and an objective evaluation framework. Through controlled error identification, it enables fine-grained diagnosis of models' analytic strengths and weaknesses across multiple dimensions. **Training Implications.** However, robust evaluation alone cannot address the fundamental methodological challenges this benchmark reveals. Cultivating genuine deep research capability requires rethinking training paradigms to reconcile heterogeneous objectives, conflicting reward signals, and multi-dimensional learning requirements.

# 4 PARALLEL RELATIVE POLICY OPTIMIZATION

## 4.1 PRELIMINARY

GRPO is a reinforcement learning algorithm designed to optimize large language models (LLMs) by leveraging group-based relative advantages. For each question $q$, GRPO samples a group of

outputs $\{o_i\}_{i=1}^{G}$ from the old policy $\pi_{\text{old}}$, where $P(Q)$ denotes the distribution over questions and $|o_i|$ represents the length of the $i$-th output sequence. The optimization process combines the probability ratio between the current and old policies with group-based relative advantages. Specifically, the probability ratio $r_{i,t}(\theta)$ quantifies how the current policy $\pi_\theta$ assigns probability to each token $o_{i,t}$ (the $t$-th token in the $i$-th sequence) compared to the old policy, while the advantage $\hat{A}_i$ is computed by normalizing the group-level rewards $\{R_i\}_{i=1}^{G}$ as follows:

$$\hat{A}_i = \frac{R_i - \bar{R}}{\sigma}, \quad r_{i,t}(\theta) = \frac{\pi_\theta(o_{i,t}|q, o_{i,<t})}{\pi_{\text{old}}(o_{i,t}|q, o_{i,<t})}, \tag{1}$$

where $R_i$ is the reward assigned to the $i$-th response, $\bar{R} = \frac{1}{G}\sum_{j=1}^{G} R_j$ is the group mean reward, and $\sigma = \sqrt{\frac{1}{G-1}\sum_{j=1}^{G}(R_j - \bar{R})^2}$ is the group standard deviation. This normalization ensures that the advantage is calculated relative to the group baseline with unit variance, while accounting for the variability within the group. GRPO optimizes the policy model $\pi_\theta$ by maximizing the following clipped objective, together with a KL penalty term:

$$J_{\text{GRPO}}(\theta) = \mathbb{E}_{q \sim P(Q), \{o_i\}_{i=1}^{G} \sim \pi_{\text{old}}(\cdot|q)} \left[ \frac{1}{G} \sum_{i=1}^{G} \frac{1}{|o_i|} \sum_{t=1}^{|o_i|} L_{\text{clip}}(r_{i,t}(\theta), \hat{A}_i) \right], \tag{2}$$

where $L_{\text{clip}}(r, A) = \min(r \cdot A, \text{clip}(r, 1 - \epsilon, 1 + \epsilon) \cdot A)$, and $\epsilon$ is the clipping hyperparameter that constrains the ratio $r_{i,t}(\theta)$ to ensure stable updates. For notational simplicity, we omit the KL divergence penalty term in this and subsequent formulations.

Achieving strong deep research capability requires models to master multiple complex dimensions simultaneously, including visual recognition, logical consistency, and numerical reasoning. However, this multi-dimensional nature introduces significant optimization challenges.

First, multi-dimensional reward interference arises when heterogeneous signals are aggregated. Existing methods such as GRPO collapse all reward dimensions into a single scalar, which compresses variability, weakens optimization signals, and diminishes the discriminative power of advantage estimation. In contrast, our proposed PRPO preserves reward information by processing each dimension independently, thereby retaining signal integrity and enabling stronger optimization guidance (More details please refer to Fiture 4 in Appendx A).

Second, data joint optimization conflicts occur because gradients from heterogeneous tasks interact unevenly: high-magnitude signals from simple tasks dominate training, while weaker but more informative signals from complex tasks are suppressed. This imbalance skews optimization toward certain data types, preventing models from fully leveraging data diversity and realizing balanced capability growth. A detailed visualization comparing GRPO and PRPO, together with extended quantitative analysis, is provided in Appendix C.

## 4.2 Reward Parallel Relative Policy Optimization

To address the limitations of GRPO in handling multi-dimensional rewards, we propose Reward-PRPO to decomposes the optimization process across different reward dimensions. Rather than aggregating multiple reward signals into a single scalar value, Reward-PRPO treats each reward dimension as an independent optimization objective. This enables the model to learn specialized capabilities while maintaining overall coherence. Specifically, for a given question $q$ and a group of generated outputs $\{o_i\}_{i=1}^{G}$, let $\{R_i^{(k)}\}_{k=1}^{K}$ denote the rewards across $K$ different dimensions for the $i$-th response. Reward-PRPO computes dimension-specific advantages for each reward component:

$$\hat{A}_i^{(k)} = \frac{R_i^{(k)} - \bar{R}^{(k)}}{\sigma^{(k)}}, \quad k = 1, 2, \ldots, K, \tag{3}$$

where $\bar{R}^{(k)}$ and $\sigma^{(k)}$ are the mean and standard deviation for the $k$-th reward dimension, respectively. This decomposition preserves the distinct optimization signals from each dimension, preventing the signal interference effects that occur in aggregated reward schemes. The Reward-PRPO objective function is formulated as a weighted combination of dimension-specific policy gradient losses:

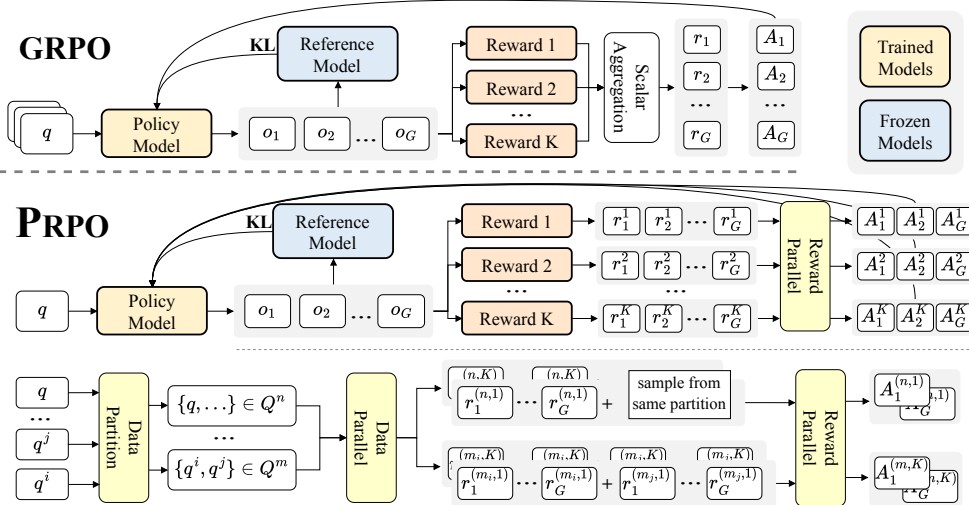

Figure 3: Demonstration of GRPO and our PRPO. PRPO unifies Reward-PRPO and Data-PRPO, partitioning data into capability-based groups and decomposing rewards across dimensions. This approach addresses multi-dimensional reward conflicts and data optimization conflicts, enabling balanced training across complex tasks.

$$J_{\text{Reward-PRPO}}(\theta) = \sum_{k=1}^{K} \lambda_k \mathbb{E}_{q \sim P(Q), \{o_i\}_{i=1}^{G} \sim \pi_{\text{old}}(\cdot|q)} \left[ \frac{1}{G} \sum_{i=1}^{G} \frac{1}{|o_i|} \sum_{t=1}^{|o_i|} L_{\text{clip}}(r_{i,t}(\theta), \hat{A}_i^{(k)}) \right], \qquad (4)$$

where $\lambda_k \geq 0$ with $\sum_{k=1}^{K} \lambda_k = 1$ represents the weight assigned to the $k$-th reward dimension, and $r_{i,t}(\theta)$ is the probability ratio as defined in GRPO. Note that we omit the KL term here, which follows the same implementation as GRPO.

## 4.3 DATA PARALLEL RELATIVE POLICY OPTIMIZATION

The original GRPO performs grouping at the rollout level within individual samples, where each question $q$ generates a group of outputs $\{o_i\}_{i=1}^{G}$ from the old policy $\pi_{\text{old}}$ with unique rollout identifiers rollout_uid$_i$. While this intra-question grouping effectively normalizes rewards relative to multiple attempts at the same task, it fundamentally fails to address optimization conflicts arising from heterogeneous data types with distinct reward characteristics.

To address this fundamental limitation, we propose Data-PRPO, which extends GRPO's rollout-level grouping mechanism to capability-based partitioning across heterogeneous data types. The core innovation lies in introducing capability-based level identifiers capability_uid that partition training samples according to their underlying cognitive requirements, such as visual understanding, logical reasoning, and data analysis. Unlike GRPO's fine-grained rollout_uid that distinguishes individual responses within the same question, capability_uid enables coarse-grained categorization based on capability dimensions, creating adaptive sub-distributions $\{P(Q^{(m)})\}_{m=1}^{M}$ with homogeneous reward patterns. Within each capability-based partition, Data-PRPO computes standardized advantages using partition-specific statistics:

$$\hat{A}_i^{(m)} = \frac{R_i - \bar{R}^{(m)}}{\sigma^{(m)}}, \qquad (5)$$

where $\bar{R}^{(m)}$ and $\sigma^{(m)}$ represent the empirical mean and standard deviation of rewards within capability partition $m$. However, this capability-based partitioning faces a critical challenge: samples with extreme advantage values can dominate gradient updates and destabilize optimization. These outliers often indicate incompatibility with their assigned capability partition's reward distribution, suggesting that the initial capability-based assignment may not be optimal for all samples. To address this challenge, Data-PRPO iteratively validates samples to ensure they fall within acceptable confidence bounds:

$$\mathcal{O}^{(t)} = \{i : |\hat{A}_i^{(t)}| > \tau\}, \tag{6}$$

where $\tau$ defines the confidence threshold for acceptable advantage values within a partition. When samples exceed this threshold, the algorithm implements a fallback mechanism that relegates these outlier samples to individual rollout-level optimization:

$$\text{partition}(i, t+1) = \begin{cases} \text{rollout\_uid}_i & \text{if } i \in \mathcal{O}^{(t)}, \\ \text{partition}(i, t) & \text{otherwise.} \end{cases} \tag{7}$$

This relegation mechanism ensures that outlier samples are optimized individually, preventing them from distorting the reward statistics of their capability partitions while preserving partition-based optimization benefits for samples that fit well within their assigned capability partitions. The iterative validation process terminates when no outliers are found ($\mathcal{O}^{(t)} = \emptyset$) or the maximum iteration limit is reached, yielding the final validated partition $\{\mathcal{G}_m\}_{m=1}^{M_{\text{final}}}$.

Upon establishing the validated data partition, Data-PRPO computes the GRPO objective function as a weighted aggregation across homogeneous partitions:

$$J_{\text{Data-PRPO}}(\theta) = \sum_{m=1}^{M_{\text{final}}} \lambda_m \mathbb{E}_{q \sim P(Q^{(m)}), \{o_i\}_{i \in \mathcal{G}_m} \sim \pi_{\text{old}}(\cdot|q)} \left[ \frac{1}{|\mathcal{G}_m|} \sum_{i \in \mathcal{G}_m} \frac{1}{|o_i|} \sum_{t=1}^{|o_i|} L_{\text{clip}}(r_{i,t}(\theta), \hat{A}_i^{(m)}) \right], \tag{8}$$

where $\lambda_m \geq 0$ with $\sum_{m=1}^{M_{\text{final}}} \lambda_m = 1$ and $\hat{A}_i^{(m)}$ represents the advantage computed within partition $\mathcal{G}_m$ using partition-specific statistics. Through this capability-based partitioning and iterative validation framework, Data-PRPO successfully resolves optimization conflicts in multi-type data training, enabling balanced development across different capability dimensions.

## 4.4 PARALLEL RELATIVE POLICY OPTIMIZATION

Building upon Reward-PRPO and Data-PRPO, we present Overall PRPO that unifies both approaches to simultaneously address multi-dimensional reward conflicts and data joint optimization conflicts.

PRPO first partitions training samples into capability-based groups using Data-PRPO, then decomposes rewards across multiple dimensions within each partition following Reward-PRPO. For samples in capability partition $m$ and reward dimension $k$, the advantage is computed as:

$$\hat{A}_i^{(k,m)} = \frac{R_i^{(k)} - \bar{R}^{(k,m)}}{\sigma^{(k,m)}} \tag{9}$$

where $\bar{R}^{(k,m)}$ and $\sigma^{(k,m)}$ are the mean and standard deviation for the $k$-th reward dimension within capability partition $m$. The unified objective function combines both parallelization strategies:

$$J_{\text{PRPO}}(\theta) = \sum_{m=1}^{M_{\text{final}}} \lambda_m \sum_{k=1}^{K} \lambda_k \mathbb{E}_{q \sim P(Q^{(m)}), \{o_i\}_{i \in \mathcal{G}_m} \sim \pi_{\text{old}}(\cdot|q)} \left[ \frac{1}{|\mathcal{G}_m|} \sum_{i \in \mathcal{G}_m} \frac{1}{|o_i|} \sum_{t=1}^{|o_i|} L_{\text{clip}}(r_{i,t}(\theta), \hat{A}_i^{(k,m)}) \right]. \tag{10}$$

This comprehensive framework effectively prevents signal interference between reward dimensions while mitigating optimization conflicts from heterogeneous data types, enabling balanced capability development across complex multi-task scenarios.

## 5 EXPERIMENTS

### 5.1 SETTINGS

**MCDR-Bench Evaluation**. We compare against both commercial and open-source models to demonstrate the effectiveness of our approach. For commercial models, we test GPT-4o (Hurst et al., 2024), GPT-4.1 (OpenAI, 2025), Claude-3.7 Sonnet(Anthropic, 2025b), and Gemini-2.5-Pro(Comanici et al., 2025) using their respective APIs. For open-source models, we evaluate ChartGemma-3B(Masry et al., 2024), Qwen2.5-VL-3B-Instruct(Bai et al., 2025), ChartInstruct-

Table 1: Performance comparison on MCDR-Bench across different splits. We evaluate both GRPO and our PRPO under two prompt settings: direct answer generation and chain-of-thought reasoning (Think). The Think setting encourages models to provide step-by-step reasoning before generating final answers, which is particularly important for complex chart analysis tasks.

| Model | BG | FE | RL | DR | F/P | Overall | Mean |
|---|---|---|---|---|---|---|---|
| GPT-4o | 27.17 | 21.89 | 41.01 | 47.45 | 60.00 | 35.83 | 38.89 |
| GPT-4.1 | 34.68 | 19.50 | 72.25 | 77.11 | 79.22 | 50.87 | 55.61 |
| Claude-3.7 Sonnet | 68.78 | 57.27 | 89.45 | 85.02 | 86.97 | 74.96 | 77.08 |
| Gemini-2.5-Pro | **81.21** | **87.31** | **91.44** | **93.78** | **92.98** | **89.29** | **89.34** |
| ChartGemma-3B | 7.53 | 5.02 | 7.51 | 9.25 | 14.29 | 8.14 | 8.29 |
| Qwen2.5-VL-3B-Instruct | 1.88 | 0.81 | 3.06 | 10.83 | 12.58 | 4.96 | 5.69 |
| ChartInstruct-LLama2-7B | 4.90 | 3.38 | 5.84 | 6.69 | 5.54 | 5.09 | 5.24 |
| LLaVA-Next-7B | 10.73 | 7.81 | 23.08 | 20.66 | 27.29 | 16.96 | 17.76 |
| InternVL2.5-8B | 21.09 | 20.77 | 37.27 | 32.67 | 47.12 | 30.64 | 31.59 |
| Llama-3.2-11B-Vision | 28.06 | 33.02 | 36.02 | 36.41 | 39.44 | 34.40 | 34.56 |
| Qwen2.5-VL-7B-Instruct | 23.35 | 39.43 | 51.04 | 37.59 | 45.63 | 40.01 | 39.51 |
| w/ GRPO | 41.24 | 51.69 | 75.38 | 66.14 | 77.40 | 61.71 | 62.26 |
| **w/ PRPO** | **50.65** | **61.38** | **81.78** | **72.83** | **84.01** | **69.62** | **69.90** |
| | *+9.41* | *+9.69* | *+6.40* | *+6.69* | *+6.61* | *+7.91* | *+7.64* |
| w/ GRPO Think | 43.13 | 48.77 | 77.75 | 70.28 | 81.02 | 63.00 | 63.99 |
| **w/ PRPO Think** | **62.90** | **65.23** | **88.87** | **80.91** | **87.21** | **76.26** | **76.89** |
| | *+19.77* | *+16.46* | *+11.12* | *+10.63* | *+6.19* | *+13.26* | *+12.90* |

Table 2: Performance comparison on ChartQAPRO across different splits.

| Model | Factoid | MCQ | Conv. | FactChk | Hypo. | Overall | Mean |
|---|---|---|---|---|---|---|---|
| GPT-4o | 35.76 | 46.72 | 34.75 | 45.49 | 28.91 | 37.67 | 38.33 |
| Claude-3.5 Sonnet | 38.84 | 51.40 | **44.53** | 55.60 | **45.48** | 43.58 | 46.57 |
| Gemini-Flash-2.0 | **43.43** | **60.28** | 40.25 | **67.62** | 24.47 | **46.85** | **47.15** |
| ChartGemma-3B | 6.86 | 0.0 | 16.00 | 1.22 | 6.53 | 6.84 | 6.12 |
| TinyChart-3B | 8.52 | 7.00 | 17.46 | 33.19 | 16.06 | 13.25 | 16.45 |
| ChartInstruct-LLama2-7B | 7.09 | 0.0 | 3.77 | 0.0 | 6.91 | 4.88 | 3.55 |
| LLaVA-Next-7B | 15.35 | 35.98 | 21.09 | 41.80 | 17.79 | 21.97 | 25.83 |
| InternVL2.5-8B | 35.21 | 25.70 | 32.26 | 53.27 | 29.61 | 35.67 | 35.29 |
| Llama-3.2-11B-Vision | 12.34 | 2.33 | 0.19 | 27.18 | 10.93 | 11.09 | 10.68 |
| Qwen2.5-VL-7B-Instruct | 27.49 | 37.85 | **55.22** | 46.72 | 44.40 | 36.31 | 41.33 |
| w/ ChartReasoner-SFT | - | - | - | - | - | 37.94 | - |
| w/ ChartReasoner-GRPO | - | - | - | - | - | 39.97 | - |
| **w/ PRPO** | **36.24** | **50.47** | 49.63 | **53.28** | **53.69** | **42.95** | **47.69** |
| | *+8.75* | *+12.62* | *-5.59* | *+6.56* | *+9.29* | *+6.64* | *+6.36* |

LLama2-7B(Han et al., 2023), LLaVA-Next-7B (Liu et al., 2024), InternVL2.5-8B(Chen et al., 2024), and Llama-3.2-11B-Vision(Grattafiori et al., 2024). We maintain consistency with the default hyperparameters and prompt methods provided by each model. To demonstrate the effectiveness of our PRPO, we specifically fine-tune Qwen2.5-VL-7B-Instruct using different optimization strategies: baseline GRPO and our proposed PRPO.

**ChartQAPRO Evaluation**. We further validate our approach on ChartQAPRO(Masry et al., 2025), a specialized benchmark for chart question answering. The model list includes commercial models GPT-4o(Hurst et al., 2024), Claude-3.5 Sonnet(Anthropic, 2025a), and Gemini-Flash-2.0(Comanici et al., 2025), as well as open-source models ChartGemma-3B, TinyChart-3B(Zhang et al., 2024a), ChartInstruct-LLama2-7B, LLaVA-Next-7B, InternVL2.5-8B, Llama-3.2-11B-Vision and ChartReasoner (Jia et al., 2025).

## 5.2 MCDR-BENCH RESULTS

Table 1 presents comprehensive evaluation results on MCDR-Bench across five capability dimensions: Background knowledge (BG), Relationship construction (RL), Factual information extraction (FE), Deep Research Report (DR), and Forecast/Plan (F/P). These dimensions correspond to the five

Table 3: Ablation results of different algorithms on MCDR-Bench.

| Method | Performance (%) |
|---|---|
| Baseline | 40.01 |
| GRPO | 61.71 |
| Reward-PRPO | 64.30 |
| Data-PRPO | 63.55 |
| PRPO | **69.62** |

Table 4: Reward Ablation Results on geometry3k.

| Method | Accuracy (%) ↑ | Response Length ↓ |
|---|---|---|
| Format Reward | 36.17 | **328** |
| Accuracy Reward | **44.46** | 350 |
| Accuracy → Format | 43.74 | **274** |
| Format → Accuracy | **44.79** | 303 |
| GRPO | 42.02 | 356 |
| PRPO | **48.75** | **319** |

stages of our PRPO framework, where we systematically inject targeted errors to transform subjective generation tasks into objective error detection tasks.

**Closed-Source Models.** Among commercial models, Gemini-2.5-Pro achieves the strongest performance with an 89.34% mean score, particularly excelling in relationship construction at 91.44% and deep research at 93.78%. Claude-3.7 follows with 77.08% mean performance, while significant gaps exist with GPT-4.1 at 55.61% and GPT-4o at 38.89%.

**Open-Source Models.** Open-source models generally underperform their commercial counterparts. Llama-3.2-11B-Vision achieves the highest mean score of 34.56%, followed by InternVL2.5-8B at 31.59%. Smaller models such as ChartGemma-3B at 8.29% and Qwen2.5-VL-3B-Instruct at 5.69% struggle across all dimensions, highlighting the critical importance of model scale for complex chart reasoning tasks.

**PRPO Effectiveness.** Our PRPO algorithm demonstrates substantial improvements over the baseline GRPO approach. For direct answer generation, PRPO achieves 69.90% mean performance versus GRPO's 62.26%, representing a 7.64% improvement, with notable gains in background knowledge of 9.41% and factual extraction of 9.69%. The improvement becomes more pronounced in the Think setting, where PRPO reaches 76.89% versus GRPO's 63.99%, achieving a 12.90% improvement, with background knowledge showing the largest gain of 19.77%. This demonstrates PRPO's effectiveness in optimizing complex reasoning tasks that require coordinated multi-dimensional capabilities. Notably, PRPO with the Think setting at 76.89% approaches Claude-3.7's performance at 77.08%, effectively bridging the gap between open-source and commercial models through reinforcement learning optimization.

## 5.3 CHARTQAPRO RESULTS

Table 2 presents evaluation results on ChartQAPRO (Masry et al., 2025), an external benchmark for chart question answering across five task types: Factoid questions, Multiple Choice Questions (MCQ), Conversational reasoning (Conv.), Fact Checking (FactChk), and Hypothetical scenarios (Hypo.). This evaluation demonstrates the generalizability of our PRPO algorithm beyond our proposed MCDR Bench.

**Baseline Models.** Among closed-source models, Gemini-Flash-2.0 achieves the best performance with a 47.15% mean score, followed by Claude-3.5 Sonnet at 46.57% and GPT-4o at 38.33%. Open-source models generally underperform, with InternVL2.5-8B leading at 35.29% and the baseline Qwen2.5-VL-7B-Instruct achieving 41.33% mean performance.

**Generalization.** Our PRPO algorithm demonstrates consistent effectiveness on this external benchmark, achieving 47.69% mean performance compared to the baseline's 41.33%, representing a 6.36% improvement. The improvement is particularly notable in MCQ with a 12.62% gain and hypothetical scenarios with a 9.29% gain, suggesting that PRPO's parallel optimization strategy effectively handles diverse reasoning requirements across different task types. These results, consistent with MCDR-Bench findings, validate both the feasibility of our objective error discrimination evaluation paradigm—which transforms difficult-to-assess open generation tasks into approximately objective assessments with unique answers—and the effectiveness of our PRPO algorithm through controlled comparisons and superior performance relative to contemporary work.

## 5.4 ABLATION STUDY

We conduct comprehensive ablation experiments to demonstrate the effectiveness of our proposed algorithm.

**Overall**. Table 3 presents the ablation results for the components of our PRPO. The improvement from the baseline model's 40.01% to GRPO's 61.71% validates the fundamental effectiveness of relative policy optimization. Reward-PRPO achieves 64.30% through multi-dimensional reward parallel optimization, while Data-PRPO attains 63.55% through data parallel grouping. The complete PRPO algorithm achieves the best performance of 69.62%, fully validating our proposed parallel optimization framework.

Table 5: Data Ablation Results.

| Data | Score |
|------|-------|
| Level1 | 34.89 |
| Level2 | 39.79 |
| Level3 | 41.02 |
| Level4 | 50.32 |
| Level5 | 32.30 |
| Low Level | 32.17 |
| High Level | 18.78 |
| Data-Parallel | **63.55** |

**Reward-PRPO**. As shown in Table 4, we conduct ablation studies on different rewards on geometry3k (Lu et al., 2021) benchmark. Different reward functions produce significantly differentiated impacts: format reward controls response length to 328 but achieves only 36.17% accuracy; accuracy reward attains 44.46% accuracy but increases response length to 350, indicating inherent conflicts between optimization dimensions. GRPO's scalar aggregation leads to signal interference, as evidenced by its 42.02% accuracy and 356 response length—different reward signals cancel each other out, resulting in mediocre performance. In contrast, PRPO avoids this interference through parallel optimization, achieving 48.75% accuracy while controlling response length to 319, representing a 6.73% improvement over GRPO.

**Data-PRPO**. Table 5 shows comprehensive ablation experiments on the composition of the training data. We validate different capability dimension data using Level1-Level5 individual training and aggregated training (Low Level: Level1-3, High Level: Level4-5). Results reveal non-linear effects: Level4 achieves the highest performance of 50.32%, while Level5 only reaches 32.30%. Simple aggregation strategies show limited performance: Low Level achieves 32.17% and High Level only 18.78%, demonstrating that traditional strategies cannot handle optpointimization conflicts in heterogeneous data. Data-PRPO's capability-based grouping achieves 63.55%, representing a 13.23% improvement over the best single combination.

## 6 CONCLUSION

We present a unified framework that systematically advances chart deep research capabilities through coordinated solutions to both training and evaluation challenges. MCDR-Bench addresses the fundamental evaluation bottleneck by leveraging the "error uniqueness principle" to transform subjective generation assessment into objective error identification, enabling scalable and quantifiable evaluation of complex analytical reasoning while substantially reducing dependence on expert annotators. PRPO tackles the training bottleneck by performing parallel optimization across reward dimensions and capability partitioning across data types, effectively disentangling conflicts between heterogeneous data and multi-dimensional reward signals that previously constrained the development of sophisticated analytical capabilities. Extensive experiments demonstrate that the synergy between our evaluation methodology and training approach systematically advances chart deep research capabilities beyond surface-level processing toward genuine analytical reasoning and strategic insight generation, as evidenced by improved exploration dynamics, enhanced optimization stability, and consistently superior performance across multiple analytical dimensions.

**Acknowledgement**: This work is supported in part by the National Natural Science Foundation of China under Grant No.62576365.

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

# A  REWARD ANALYSIS

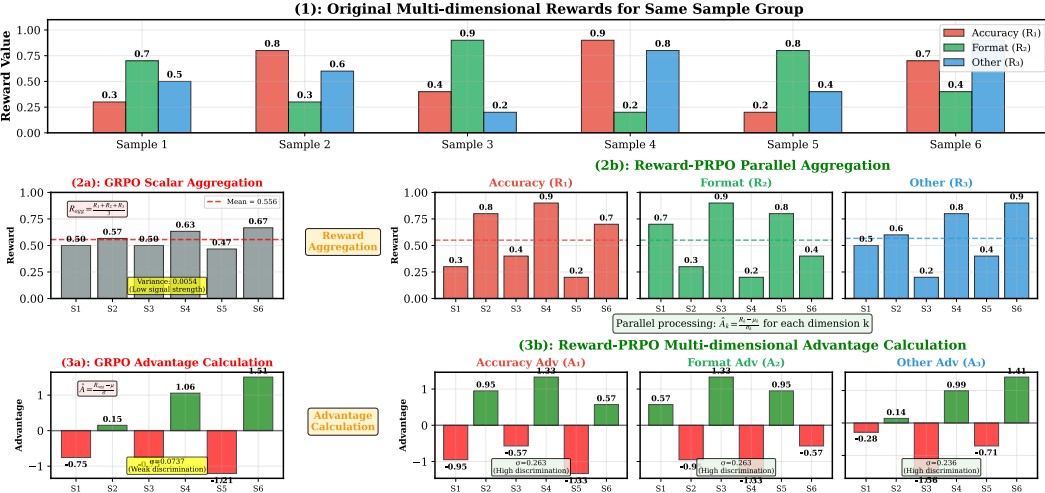

Figure 4: Comparison of multi-dimensional reward processing between GRPO and PRPO. (1) Original multi-dimensional rewards for the same sample. (2a) GRPO aggregates rewards into a single scalar value, resulting in low variance and weak signal strength, which diminishes the distinct optimization signals from each reward dimension. (3a) GRPO's advantage calculation further exhibits weak discrimination due to the loss of variability in aggregated rewards. (2b) PRPO independently processes each reward dimension, preserving signal integrity. (3b) PRPO's dimension-specific advantage calculation achieves higher discrimination, effectively capturing the unique contributions of each reward type and enabling more effective optimization.

# B  IMPLEMENTATION DETAILS

We filter and collect high-quality chart data from SophiaVL-R1-130k (Fan et al., 2025) and R1-Onevision (Yang et al., 2025). We subsequently employ GPT-4o to annotate capability dimensions for each data sample according to the five capability dimensions of chart deep research. Since not all data samples can be perfectly categorized into specific dimensions, we filter out samples that fall outside the scope or exhibit low quality. Ultimately, we construct a high-quality RL dataset of 10k samples with relatively balanced distribution across all capability dimensions. In our PRPO, we introduce two hyperparameters, $\lambda_k$ and $\lambda_m$. The hyperparameter $\lambda_k$ corresponds to the mean of the reward dimension, serving as a scaling factor to normalize rewards across different capability dimensions. The hyperparameter $\lambda_m$ represents the number of data groups, controlling how samples are partitioned for group-wise relative reward computation. We maintain the default hyperparameter configuration that performs mean aggregation across reward and data group dimensions.

# C  OPTIMIZATION ANALYSIS

As shown in Figure 4, we conduct comprehensive experimental analyses of optimization conflicts on general math reasoning datasets geometry3k (Lu et al., 2021) to investigate the impact of different reward function strategies on model optimization effectiveness. The experiments employed two models of varying scales—Qwen2.5-VL-3B and Qwen2.5-VL-7B—and performed in-depth analyses of reward value and response length distributions during the training process. Five distinct training strategies were implemented to evaluate the influence of reward functions:

- **Base** (the original GRPO method utilizing scalar summation of accuracy and format rewards for advantage computation)
- **Accuracy** (employing solely the accuracy reward function)
- **Format** (utilizing exclusively the format reward function)

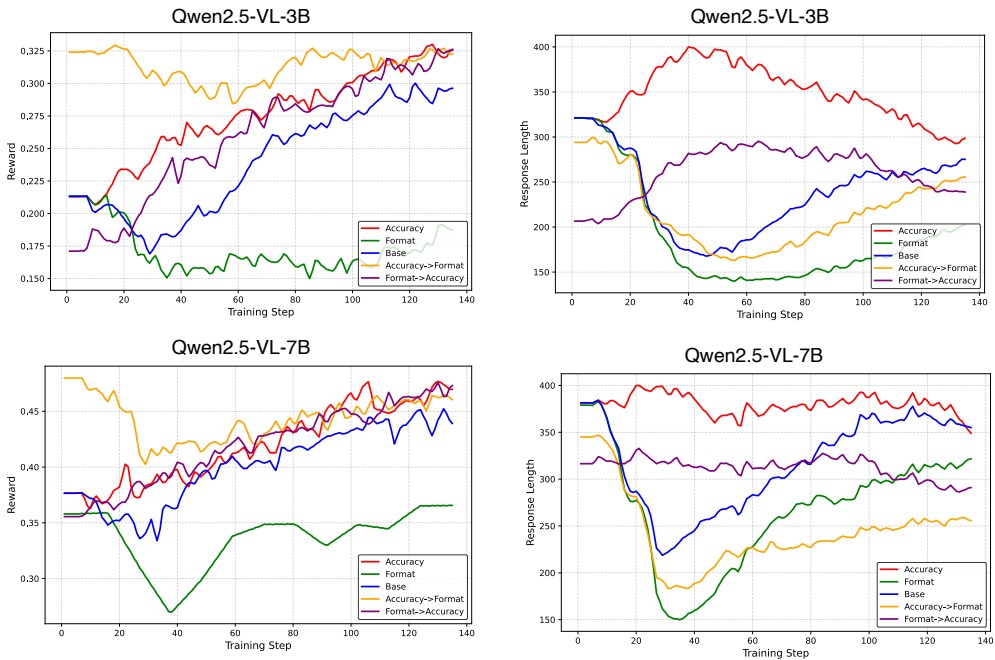

Figure 5: Comparison of reward values (left) and response lengths (right) during training for Qwen2.5-VL-3B (top) and Qwen2.5-VL-7B (bottom) models under different reward strategies.

- **Accuracy→Format** (sequential training with accuracy reward followed by format reward)

- **Format→Accuracy** (sequential training with format reward followed by accuracy reward).

Through these configurations, we analyzed the impact of different reward strategies on model optimization effectiveness and explored the unique optimization preferences of reward functions and their effects on model capabilities.

**Upper left** illustrates the reward value trajectories for the Qwen2.5-VL-3B model under different reward strategies. The results demonstrate that individual application of accuracy rewards (red curve) and format rewards (yellow curve) significantly outperforms the Base strategy (blue curve). Notably, the accuracy reward strategy (red curve) maintains consistently high reward values throughout training, indicating its distinctive optimization preference. Conversely, the Base strategy employing scalar aggregation (blue curve) exhibits lower reward values during training, suggesting that simple scalar aggregation fails to fully exploit the optimization potential of individual reward functions. Furthermore, sequential reward function strategies (purple and green curves) demonstrate superior optimization performance, particularly the Format→Accuracy strategy (purple curve), which shows pronounced advantages in late-stage reward value improvement. This indicates that isolating reward function influences enables better model capability optimization compared to unified processing approaches.

**Upper right** demonstrates the impact of different reward strategies on response length for the Qwen2.5-VL-3B model. Notably, individual accuracy reward application (red curve) leads to significant response length increases, while format reward application (yellow curve) results in substantial response length reductions. This reflects distinct generative behavior preferences of reward functions: accuracy rewards favor longer responses to enhance information coverage, whereas format rewards promote concise responses to satisfy formatting requirements. Sequential reward function strategies (purple and green curves) exhibit balanced response length trends, particularly the Format→Accuracy strategy (purple curve), which effectively controls response length in later stages while maintaining high reward values. In contrast, the Base strategy (blue curve) demonstrates considerable response length volatility, indicating that scalar aggregation cannot effectively coordinate conflicts between reward functions.

**Lower left** presents reward value trajectories for the Qwen2.5-VL-7B model under different reward strategies. Similar to the 3B model, individual application of accuracy rewards (red curve) and format rewards (yellow curve) outperforms the Base strategy (blue curve). Remarkably, sequential reward function strategies (purple and green curves) demonstrate more pronounced performance on the 7B model, particularly the Accuracy→Format strategy (green curve), which surpasses individual reward function strategies in late-stage reward value improvement. This further validates the unique optimization preferences of reward functions and demonstrates that sequential reward function application better adapts to dynamic changes in model capabilities. Conversely, the Base strategy (blue curve) exhibits consistently low reward values throughout training, confirming that scalar aggregation cannot fully utilize the unique optimization capabilities of reward functions.

**Low right** illustrates the impact of different reward strategies on response length for the Qwen2.5-VL-7B model. Consistent with the 3B model trends, individual accuracy reward application (red curve) significantly increases response length, while format reward application (yellow curve) substantially reduces response length. This further validates the distinct optimization preferences of reward functions for model generative behavior: accuracy rewards favor longer responses for comprehensive information coverage, while format rewards promote concise responses to meet formatting requirements. Sequential reward function strategies (purple and green curves) demonstrate superior balance in response length, particularly the Format→Accuracy strategy (purple curve), which effectively controls response length in later stages while maintaining high reward values. This indicates that sequential reward function application achieves compromise between different reward preferences, optimizing model generative behavior. In contrast, the Base strategy (blue curve) exhibits substantial response length volatility, confirming that scalar aggregation cannot effectively coordinate conflicts between reward functions, resulting in unstable model generative behavior.

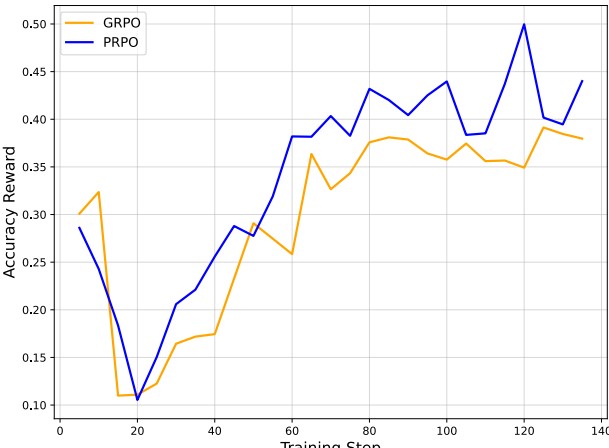

Figure 6: Accuracy reward trajectories during training for different optimization strategies. The orange curve represents GRPO, while the blue curve represents our PRPO. PRPO demonstrates faster growth in early training, stable improvement in the middle phase, and higher peak rewards in later stages, consistently outperforming GRPO. This indicates that PRPO's parallel optimization across reward dimensions and data types effectively mitigates reward sparsity while enhancing exploration capability and optimization efficiency.

## D PERFORMANCE ANALYSIS

Figure 6 compares the optimization performance of GRPO and PRPO during training, with training steps on the x-axis and accuracy reward values on the y-axis. PRPO consistently outperforms GRPO throughout the training process. Specifically, PRPO demonstrates rapid reward growth in the early stage (0-40 steps), significantly exceeding GRPO; maintains stable improvement in the middle stage (40-100 steps), exhibiting superior optimization stability; and achieves higher peak values in the late stage (>100 steps), further widening the performance gap. This advantage stems from PRPO's parallel optimization strategy across reward dimensions and data types, which effectively mitigates reward sparsity while fully exploiting multi-dimensional reward signals, thereby enhanc-

ing exploration capability and optimization efficiency. In contrast, GRPO exhibits greater volatility and consistently lower reward values, indicating weaker adaptability in sparse reward environments.

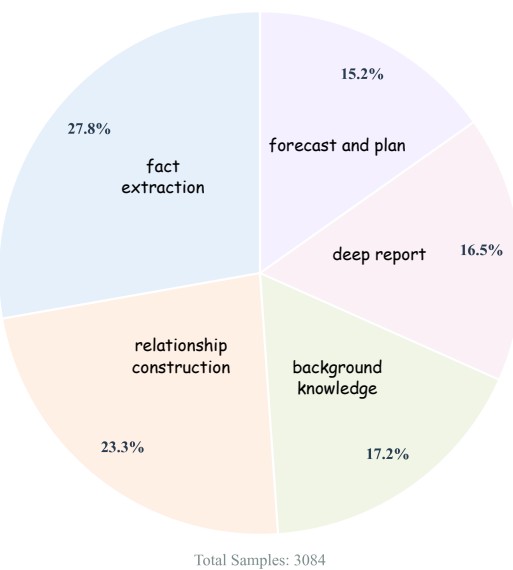

Figure 7: The pie chart illustrates the proportional distribution of error types across the five evaluation stages in the MCDR-Bench dataset, ensuring comprehensive coverage of diverse capabilities from background knowledge integration to strategic forecasting and planning.

## E  MCDR-BENCH

**From Subjective Generation to Objective Detection**. To resolve the "subjective answer diversity" dilemma, we innovatively transform traditional subjective generation evaluation into objective error identification tasks based on the "error uniqueness principle." This paradigm shift achieves three advantages: transforming unevaluable subjective tasks into objectively assessable discriminative tasks; significantly reducing dependence on professional annotators; and providing more fine-grained capability assessment. We design targeted error injection mechanisms for each of the five construction evaluation dimensions: ❶ *Background Knowledge* category injects outdated information, false backgrounds, and domain confusion errors to test external knowledge integration capabilities; ❷ *Fact Extraction* category introduces numerical reading, temporal relationship, and unit conversion errors to assess basic data understanding abilities; ❸ *Relationship Construction* category sets causal inversion, correlation strength, and hierarchical structure errors to evaluate complex relationship understanding capabilities; ❹ *Deep Report* category includes trend judgment, anomaly detection, and logical consistency errors to examine comprehensive analysis abilities; ❺ *Forecast and Plan* category covers feasibility assessment, risk evaluation, and priority ranking errors to test practical application judgment. This dimension-wise error injection strategy achieves complete capability coverage from basic data reading to advanced decision recommendations, complete difficulty gradients from simple single-point errors to complex systematic errors, ensuring ecological validity based on real deep research scenarios.

The error injection mechanism effectively evaluates deep research capabilities through its systematic capability decomposition approach. By implementing hierarchical error injection across five core evaluation dimensions, we can precisely identify model performance bottlenecks throughout the complete capability spectrum from fundamental data comprehension to advanced decision-making reasoning. This multi-dimensional error configuration encompasses both cognitive hierarchies ranging from simple fact extraction to complex relational inference, and analytical depths spanning from local details to global insights, ensuring that evaluation results comprehensively reflect models' integrated performance and practical application potential in authentic deep research scenarios.

# Background Knowledge Error

**DCF Valuation**
*All Figures in US$ m; Except Per Share Data*
*Financial Year Ended 31st Dec*
**Model Check : TRUE**                                    Index

| | | FY 2018 Actual | FY 2019 Actual | FY 2020 Actual | FY 2021 Forecast | FY 2022 Forecast | FY 2023 Forecast | FY 2024 Forecast | FY 2025 Forecast |
|---|---|---|---|---|---|---|---|---|---|
| **NPV Summary** | | | | | | | | | |
| NPV of Business | US$ m | 29,784 | 29,729 | 28,707 | 26,951 | 25,735 | 21,573 | 18,568 | 8,665 |
| NPV of Business (Terminal Value) | US$ m | 201,661 | 214,937 | 229,087 | 244,169 | 260,243 | 277,376 | 295,636 | 315,099 |
| **Corporate NPV at Beginning of the Period** | US$ m | 231,445 | 244,667 | 257,794 | 271,120 | 285,978 | 298,949 | 314,205 | 323,764 |
| Cash (Debt) at Beginning of the Period | US$ m | | (781) | 357 | 1,695 | 9,073 | 18,525 | 27,324 | 39,685 |
| **Shareholders' NPV** | US$ m | | 243,886 | 258,151 | 272,815 | 295,051 | 317,474 | 341,529 | 363,449 |
| Number of Shares Outstanding (Year End) - Adjusted | m | 850 | 850 | 850 | 850 | 850 | 850 | 850 | 850 |
| **NPV / Share** | US$ / Share | - | 286.9 | 303.7 | 321.0 | 347.1 | 373.5 | 401.8 | 427.6 |
| Growth (%) | % | | | 5.8% | 5.7% | 8.2% | 7.6% | 7.6% | 6.4% |
| Implied PER | x | | 25.79 | 25.73 | 23.31 | 30.89 | 28.26 | 27.08 | 25.70 |
| **NPV of Business** | | | | | | | | | |
| EBIT | US$ m | 11,800 | 11,428 | 13,272 | 11,652 | 13,950 | 15,790 | 17,778 | 19,955 |
| Tax Paid | US$ m | (3,031) | (2,003) | (2,096) | (2,906) | (3,468) | (3,927) | (4,436) | (4,991) |
| Depreciation & Amortisation | US$ m | 896 | 1,084 | 1,220 | 1,320 | 1,318 | 1,366 | 1,408 | 1,447 |
| **Gross Cash Flow to Firm** | US$ m | 9,664 | 10,509 | 12,396 | 10,065 | 11,801 | 13,228 | 14,750 | 16,411 |
| Capital Expenditure | US$ m | (2,949) | (2,105) | (1,722) | (3,000) | (3,000) | (3,000) | (3,000) | (3,000) |
| Change in Non - Cash Working Capital | US$ m | - | (275) | (1,100) | (258) | 285 | (1,547) | 754 | (1,727) |
| **Free Cash Flow to Firm (FCFF)** | US$ m | 6,715 | 8,130 | 9,574 | 6,808 | 9,086 | 8,681 | 12,504 | 11,684 |
| Growth (YoY) | % | | 21.1% | 17.8% | -28.9% | 33.5% | -4.5% | 44.0% | -6.6% |
| WACC | % | 6.6% | 6.6% | 6.6% | 6.6% | 6.6% | 6.6% | 6.6% | 6.6% |
| **NPV of Business at Beginning of the Period** | US$ m | 29,784 | 29,729 | 28,707 | 26,951 | 25,735 | 21,573 | 18,568 | 8,665 |
| **NPV of Business (Terminal Value)** | | | | | | | | | |
| Terminal Period Growth Rate | % | 3.0% | | | | | | | |
| Free Cash Flow to Firm (Terminal Year) | US$ m | 12,034 | | | | | | | |
| WACC | % | 6.6% | 6.6% | 6.6% | 6.6% | 6.6% | 6.6% | 6.6% | 6.6% |
| (WACC-Growth Rate) | % | 3.6% | 3.6% | 3.6% | 3.6% | 3.6% | 3.6% | 3.6% | 3.6% |
| Terminal Value | US$ m | 335,843 | 335,843 | 335,843 | 335,843 | 335,843 | 335,843 | 335,843 | 335,843 |

◄ ► … | Income Statement | Balance Sheet | Cash Flow Statement | Valuation Ratios | **DCF** | Compar … ⊕ ⁝ ◄

**Correct Statement:** The deep research report reveals that over 90% of the business's $323.8 billion forecasted NPV in FY 2025 is driven by terminal value—calculated using a 3% perpetual growth rate and $12.0 billion terminal year FCFF—making the company's intrinsic valuation highly sensitive to long-term growth and discount rate assumptions, while robust free cash flow recovery and a shift from net debt to $39.7 billion net cash underpin a 49% increase in shareholder NPV and a 6.8% CAGR in per-share value, highlighting both the strength of operational performance and the critical importance of prudent, realistic long-term planning.

**After Error Injecting:** The deep research report reveals that over 90% of the business's $323.8 billion forecasted NPV in FY 2025 is driven by terminal value—*calculated using a 3% perpetual growth rate and $12.0 billion terminal year FCFF based on global industry averages*—making the company's intrinsic valuation highly sensitive to long-term growth and discount rate assumptions, while robust free cash flow recovery and a shift from net debt to $39.7 billion net cash underpin a 49% increase in shareholder NPV and a 6.8% CAGR in per-share value, highlighting both the strength of operational performance and the critical importance of prudent, realistic long-term planning.

Figure 8: Illustration of a background knowledge error case from the MCDR-Bench.

**Error Type Distribution in MCDR-Bench**. The MCDR-Bench dataset establishes a comprehensive evaluation framework through five progressive capability assessments, as illustrated in Figure 7. The dataset consists of 3,084 samples, systematically distributed across five error types corresponding to the evaluation dimensions. The *Background Knowledge* category accounts for 17.2% of the error types, designed to test models' ability to integrate external domain-specific information and contextualize chart data within broader knowledge frameworks through queries requiring understanding beyond visual elements. The largest proportion of error types (27.8%) is allocated to the *Fact Extraction* dimension, which represents the fundamental capability of multimodal chart deep research. This category focuses on assessing models' proficiency in accurately extracting visual factual information, such as numerical values, temporal sequences, and categorical data, from complex graphical representations. The *Relationship Construction* dimension comprises 23.3% of the error types, targeting models' understanding of interconnections between elements or across multiple charts, including identifying patterns, correlations, and causal relationships. The *Deep Report* category contributes 16.5% of the error types, rigorously examining advanced analytical reasoning capabilities, such as data synthesis, report summarization, and insight distillation. Finally, the *Forecast and Plan* dimension accounts for 15.2% of the error types, emphasizing predictive and strategic

# Fact Extraction Error

**Correct Statement:** Such as the mid-2014 spike in rekeying and key transitions, which saw monthly rekeying events more than double the average (peaking at 11–12 per month versus a 5.08 monthly average) and key transitions reaching 60 per month—create a cascading strain on the entire security records and facilities management system for A.C. Moore Arts & Crafts: these surges overwhelm process capacity, leading to severe bottlenecks in early workflow steps (with over 57% of orders stalled in Steps 1 and 2), a sharp increase in orders taking 3+ days to complete (up to 43 in July 2014), and a corresponding rise in labor and loss hardware orders (the latter constituting 44% of all orders), particularly among the 45.7% of locations that are inactive—revealing that the system's efficiency and cost control under normal conditions are undermined by its vulnerability to high-volume events, process inefficiencies, and asset management gaps, all of which are tightly interlinked and require systemic, data-driven interventions to ensure resilience and sustained operational performance.

**After Error Injecting:** Such as the mid-2014 spike in rekeying and key transitions, which saw monthly rekeying events more than triple the average (*peaking at 16 per month* versus a 5.08 monthly average) and key transitions reaching 60 per month—create a cascading strain on the entire security records and facilities management system for A.C. Moore Arts & Crafts: these surges overwhelm process capacity, leading to severe bottlenecks in early workflow steps (with over 57% of orders stalled in Steps 1 and 2), a sharp increase in orders taking 3+ days to complete (up to 43 in July 2014), and a corresponding rise in labor and loss hardware orders (the latter constituting 44% of all orders), particularly among the 45.7% of locations that are inactive—revealing that the system's efficiency and cost control under normal conditions are undermined by its vulnerability to high-volume events, process inefficiencies, and asset management gaps, all of which are tightly interlinked and require systemic, data-driven interventions to ensure resilience and sustained operational performance.

Figure 9: Illustration of a fact extraction error case from the MCDR-Bench.

planning abilities, where models are challenged to demonstrate decision-making competencies in practical applications informed by chart analysis. This balanced distribution ensures comprehensive coverage of diverse error types, enabling robust evaluation of model capabilities across all dimensions of multimodal chart deep research.

# Relationship Construction Error

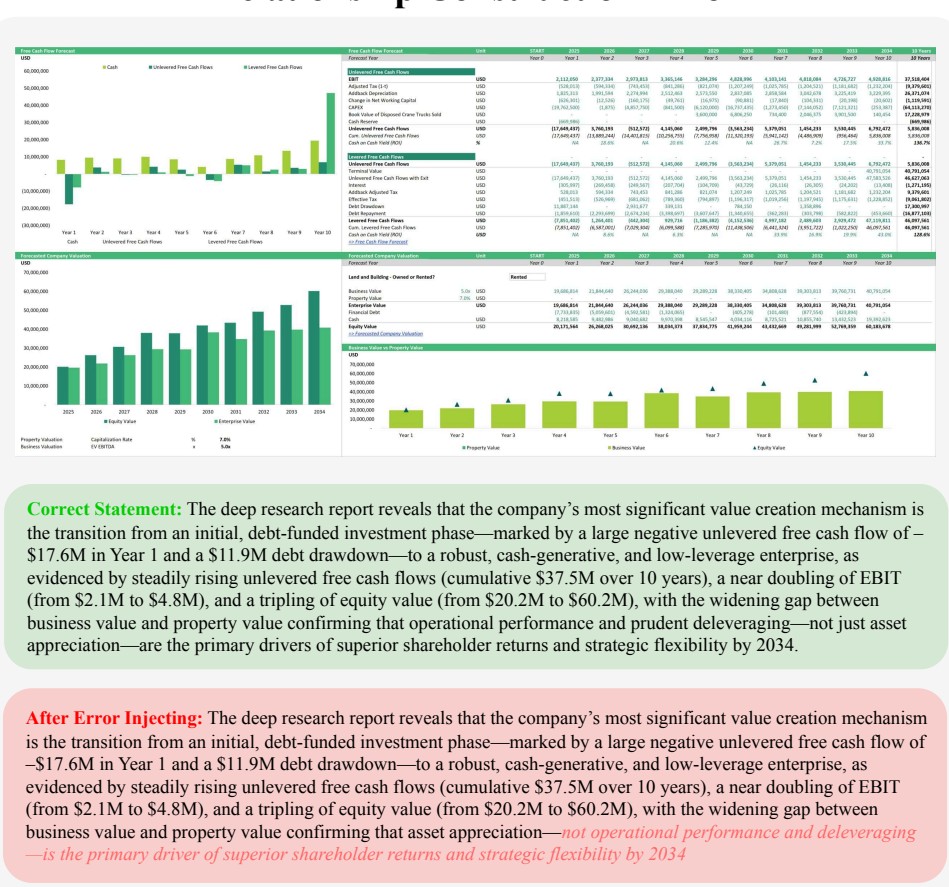

**Correct Statement:** The deep research report reveals that the company's most significant value creation mechanism is the transition from an initial, debt-funded investment phase—marked by a large negative unlevered free cash flow of – $17.6M in Year 1 and a $11.9M debt drawdown—to a robust, cash-generative, and low-leverage enterprise, as evidenced by steadily rising unlevered free cash flows (cumulative $37.5M over 10 years), a near doubling of EBIT (from $2.1M to $4.8M), and a tripling of equity value (from $20.2M to $60.2M), with the widening gap between business value and property value confirming that operational performance and prudent deleveraging—not just asset appreciation—are the primary drivers of superior shareholder returns and strategic flexibility by 2034.

**After Error Injecting:** The deep research report reveals that the company's most significant value creation mechanism is the transition from an initial, debt-funded investment phase—marked by a large negative unlevered free cash flow of –$17.6M in Year 1 and a $11.9M debt drawdown—to a robust, cash-generative, and low-leverage enterprise, as evidenced by steadily rising unlevered free cash flows (cumulative $37.5M over 10 years), a near doubling of EBIT (from $2.1M to $4.8M), and a tripling of equity value (from $20.2M to $60.2M), with the widening gap between business value and property value confirming that asset appreciation—*not operational performance and deleveraging —is the primary driver of superior shareholder returns and strategic flexibility by 2034*

Figure 10: Illustration of a relationship constriction error case from the MCDR-Bench.

## F VISUALIZATIONS

Based on the error injection evaluation mechanism of MCDR-Bench, we construct a comprehensive set of visualization cases to demonstrate the error configurations across five dimensions in our benchmark. Each case extracts correct statements from authentic deep research reports as baselines, then systematically injects targeted error types. All errors are meticulously designed and manually verified to ensure they reflect typical cognitive biases and logical fallacies encountered in real reasoning scenarios.

**Background Knowledge Error**. Figure 8 demonstrates external knowledge integration errors in financial valuation models. The correct statement indicates that over 90% of business's 323.8 billion projected NPV in 2025 is driven by terminal value, calculated based on 312.0 billion terminal free cash flow to firm (FCFF). This calculation methodology renders the company's intrinsic valuation highly sensitive to long-term growth and discount rate assumptions, while robust free cash flow recovery and net debt reduction to 39.7 billion support 49% growth in shareholder NPV and 6.8% compound annual growth rate (CAGR) in per-share value. The error injection corrupts the terminal value calculation basis to "global industry average 3% perpetual growth rate and 12.0 billion terminal FCFF," systematically undermining the original valuation model's logical foundation and numerical consistency. This error type is designed to assess large language models' capability for domain-specific knowledge accuracy identification, cross-domain knowledge transfer rationality

judgment, and knowledge integration consistency verification within complex financial modeling contexts.

# Deep Report Error

**Correct Statement:** ABC Ship Management Ltd's projected revenue growth from $6.4M in 2024 to $20.2M in 2028 (CAGR: 33.6%) is fundamentally driven by aggressive and diversified fleet expansion—particularly in container ships and bulk carriers, which together account for 70% of average revenue—while disciplined cost management and a conservative capital structure (debt-to-equity ratio of 1.2%) enable the company to sustain high profitability (gross margin ~35%, net margin rising from 7% to 9%) and exceptional returns on assets (33%) and equity (42%), creating a self-reinforcing cycle of operational scale, financial strength, and risk mitigation that positions the company for resilient, scalable, and sustainable value creation.

**After Error Injecting:** ABC Ship Management Ltd's projected revenue growth from $6.4M in 2024 to $20.2M in 2028 (CAGR: 33.6%) is fundamentally driven by aggressive and diversified fleet expansion—particularly in container ships and bulk carriers, which together account for 70% of average revenue—while disciplined cost management and a conservative capital structure (debt-to-equity ratio of 1.2%) ensure *that profitability (gross margin ~35%, net margin rising from 7% to 9%) remains stable even if operational scale were to stagnate, with high returns on assets (33%) and equity (42%) maintained independently of further fleet growth, thereby positioning the company for resilient, scalable, and sustainable value creation.*

Figure 11: Illustration of a deep report error case from the MCDR-Bench.

**Fact Extraction Error.** Figure 9 focuses on numerical information processing precision assessment. The correct statement provides detailed description of A.C. Moore Arts & Crafts security system anomalies in mid-2014: re-keying events surged from normal 5.08 to 11-12 occurrences, key conversion events reached 60, resulting in 57% of orders stagnating in early workflow steps, order completion times exceeding 3 days peaked at 43 instances in July, and hardware order losses constituted 44% of all orders. The error injection misrepresents the re-keying event peak as 16 occurrences. While the numerical deviation appears minimal, within systematic analysis frameworks it could lead to significant misjudgments in security risk assessment and resource allocation decisions. This error design aims to examine large language models' numerical information extraction accuracy, temporal data comprehension capabilities, and logical consistency maintenance abilities in multi-dimensional data cross-validation scenarios.

## Forecast and Plan Error

**Correct Statement:** Despite Germany achieving a significant milestone with 47.0% of its population (39.1 million people) having received at least one COVID-19 vaccine dose and 19.9 million fully vaccinated, the deep research report reveals that persistent regional disparities—ranging from 51.2% coverage in Bremen to just 42.1% in Sachsen—pose a critical challenge to national herd immunity; the recent plateau in daily vaccinations (after peaking at over 1.2 million doses/day in May) further underscores that without targeted, data-driven interventions in lagging regions, Germany risks both prolonging the pandemic and facing localized outbreaks, making the elimination of these regional gaps the most essential lever for accelerating national progress and safeguarding public health.

**After Error Injecting:** Despite Germany achieving a significant milestone with 47.0% of its population (39.1 million people) having received at least one COVID-19 vaccine dose and 19.9 million fully vaccinated, the deep research report reveals that persistent regional disparities—ranging from 51.2% coverage in Bremen to just 42.1% in Sachsen—pose a critical challenge to national herd immunity; however, the recent plateau in daily vaccinations (after peaking at over 1.2 million doses/day in May) *is expected to resolve naturally as vaccine supply increases, making broad, untargeted national efforts sufficient for accelerating progress and safeguarding public health.*

Figure 12: Illustration of a forecast and plan error case from the MCDR-Bench.

**Relationship Construction Error**. Figure 10 examines high-order cognitive capabilities in causal reasoning and logical relationship identification. The correct statement constructs a complex value creation mechanism: strategic transformation from debt-driven investment phase to cash-generating, low-leverage enterprise, supported by quantitative indicators including unlevered free cash flow growth from -17.6 million to 37.5 million, EBIT expansion from 2.1 million to 4.8 million, and equity value increase from 20.2 million to 60.2 million, explicitly identifying operational performance and prudent deleveraging (rather than asset appreciation) as primary drivers of superior shareholder returns. The error injection inverts this core causal relationship, erroneously identifying asset appreciation as the primary driver. This error type is designed to evaluate large language models' multi-level causal relationship identification capabilities, structured understanding of complex business logic, and key driving factor discrimination abilities within multi-variable interactive systems.

**Deep Report Error**. Figure 11 examines systematic analysis and logical consistency comprehensive judgment capabilities. The correct statement constructs a complete business model analysis framework for ABC Ship Management Ltd.: revenue growth from 6.4 million to 20.2 million (CAGR 33.6%) driven by diversified fleet expansion in container and bulk carriers (comprising 70% of average revenue), maintaining high profitability through rigorous cost management and conservative capital structure (debt-to-equity ratio 1.2%), achieving exceptional return on assets (33%) and return on equity (42%), forming a self-reinforcing cycle of operational scale, financial strength, and risk mitigation. The error injection asserts that profitability can remain stable despite operational scale stagnation, and that asset and equity returns are independent of fleet growth, systematically disrupting the original logical closed-loop structure. This error type aims to examine large language models' systematic thinking capabilities, multi-dimensional business logic integration analysis abilities, and intrinsic correlation identification judgment capabilities within complex business ecosystems.

**Forecast and Plan Error**. Figure 12 evaluates policy analysis and strategic planning practical application judgment capabilities. The correct statement, based on comprehensive COVID-19 vaccination data in Germany (47.0% total population vaccinated, regional coverage disparities from Bremen's 51.2% to Saxony's 42.1%, post-peak stagnation following May's daily vaccination rates exceeding 1.2 million doses), constructs a systematic public health risk assessment framework, emphasizing regional disparities' challenges to herd immunity and the urgency of data-driven intervention measures. The error injection claims vaccination stagnation will naturally resolve with increased vaccine supply without targeted interventions, fundamentally misjudging the structural nature of the problem and the necessity of solutions. This error type is designed to assess large language models' policy analysis reasoning capabilities, systematic thinking abilities in multi-level risk assessment, and intervention strategy rationality judgment capabilities within complex social systems.

Our error injection strategy achieves complete capability spectrum coverage from basic information processing to high-order cognitive reasoning, constructs multi-dimensional difficulty gradients from localized data deviations to systematic logical deficiencies, ensuring ecological validity and scientific rigor in assessment based on authentic deep research scenarios.

## G  QUALITATIVE ANALYSIS

As shown in Figure 13, to comprehensively evaluate the performance improvements achieved through PRPO reinforcement learning training, we conducted a detailed qualitative analysis comparing the base model Qwen2.5VL-7B with our PRPO-trained model on multimodal chart deep research tasks. The consistent improvements not only demonstrate the effectiveness of our PRPO algorithm, but also validate that our MCDR-Bench achieves effective comprehensive evaluation of models' deep research and report generation capabilities.

### G.1  DEEP RESEARCH CAPABILITY

In the financial performance analysis (     block), the PRPO-trained model demonstrated superior analytical depth. Specifically, the our model supplemented the analysis with critical quantitative information, including the number of stores (306) and product categories (1,690), providing enhanced granularity and credibility to the financial overview, while the base model omitted these essential contextual details. Regarding seasonal trend identification, although both models recognized seasonal patterns, the our model further specified a "further decline" in September and October, delivering more precise temporal analysis.

In the channel performance analysis dimension (     block), both models exhibited high similarity, providing identical data insights without significant differentiation. However, it is noteworthy that neither model delved into profitability analysis across different channels, representing a potential opportunity for enhanced analytical depth.

The product category performance analysis (     block) revealed significant advantages of the our model. The our model provided more detailed product category breakdowns, encompassing additional categories such as monitors, smartphones, and touch screen phones, which the base model overlooked. In terms of actionable insights, the our model specifically emphasized the growth po-

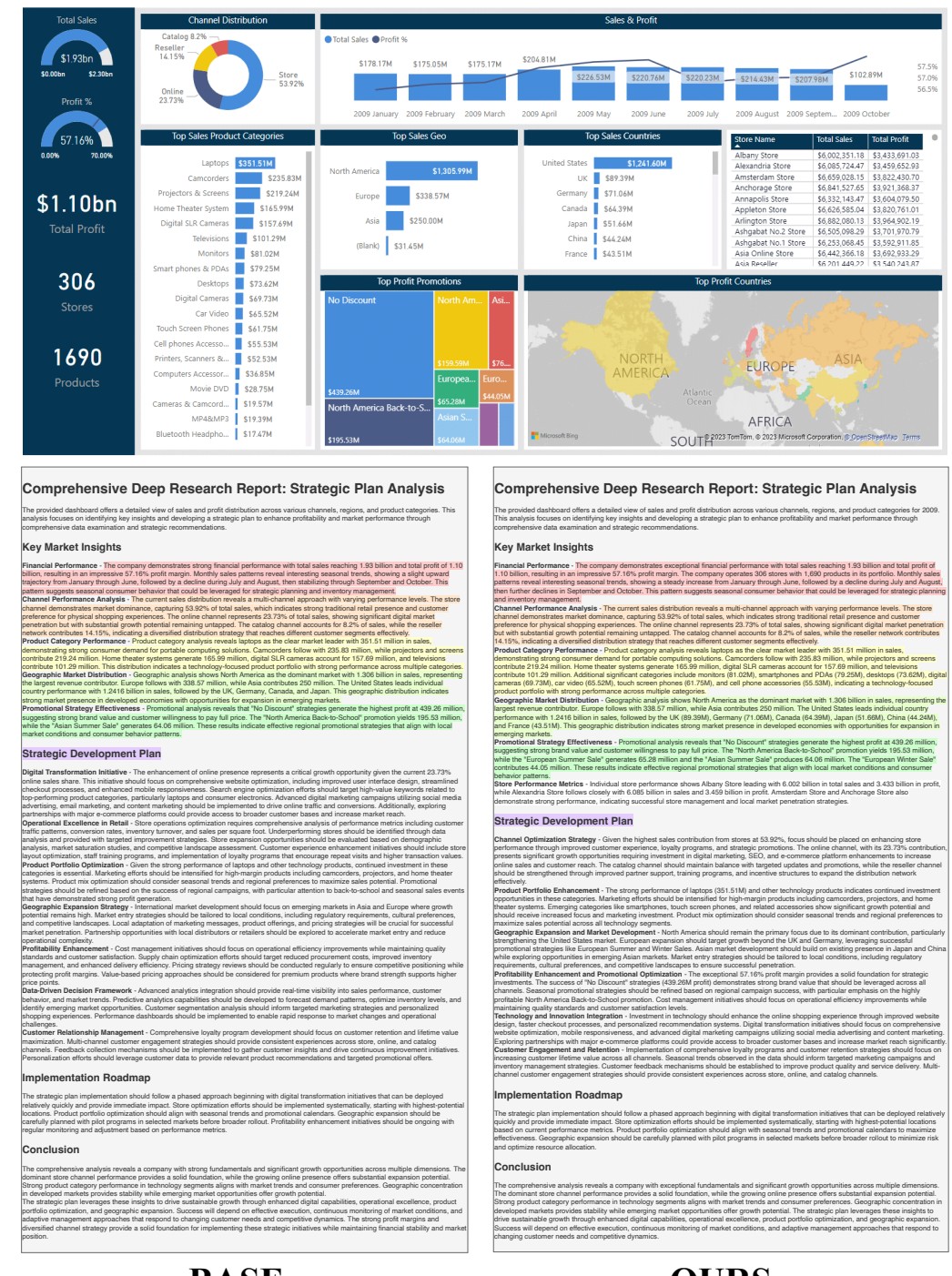

Figure 13: Qualitative Results.

tential of emerging categories like smartphones and accessories, offering forward-looking market opportunity identification, while the base model focused solely on top-performing traditional categories.

Geographic market distribution analysis (     block) further highlighted the our model's advantages in data granularity. The our model provided specific country-level sales figures (e.g., UK: $89.39M,

Germany: $71.06M), while the base model lacked such precise geographic segmentation. In strategic recommendations, the our model explicitly identified expansion opportunities in emerging markets, whereas the base model failed to elaborate on specific strategies for these markets.

Promotional strategy effectiveness evaluation (█ block) demonstrated the our model's more comprehensive promotional campaign coverage. The our model included analysis of additional promotional activities, such as the "European Summer Sale" ($65.28M) and "European Winter Sale" ($44.05M), which were omitted in the base model. Regarding actionable recommendations, the our model proposed refined seasonal promotional strategies based on regional success patterns, while the base model provided no specific optimization suggestions.

### G.2 FORECAST AND PLAN

In the future forecasting and strategic decision-making section (█ block), both models exhibited varying depths across multiple strategic dimensions. For digital transformation, while the base model focused on website optimization and e-commerce partnerships, the our model introduced "personalized recommendation systems," reflecting a more advanced technological approach. In store optimization, the base model emphasized operational metrics like inventory turnover, whereas the our model prioritized customer experience enhancement.

our model demonstrated superior comprehensiveness in product portfolio strategy by incorporating emerging categories like smartphones and accessories, while the base model focused solely on traditional high-margin products. For geographic expansion, the ours enhanced the base framework by integrating specific promotional strategies such as European seasonal sales campaigns, making expansion plans more actionable and data-driven.

Regarding profitability enhancement, the our model added promotional optimization dimensions, emphasizing "No Discount" strategies and seasonal campaign refinements, demonstrating a more dynamic approach compared to the base model's focus on cost management and supply chain optimization. In customer engagement, the our model incorporated multi-channel strategies ensuring consistent experiences across all touch points, while the base model provided only general loyalty program suggestions.

Comprehensive analysis revealed that our PRPO-trained model consistently provided more detailed data support, stronger linguistic expressions, and more specific actionable recommendations. These results demonstrate that PRPO reinforcement learning training significantly enhanced model performance in complex multimodal chart deep research tasks, particularly in data integration, insight depth, and the action ability of strategic recommendations for report analysis and generation.

## H ETHICS STATEMENT

While our research advances multimodal chart deep research capabilities, we acknowledge potential risks including the possibility of automated systems making erroneous analytical conclusions that could mislead decision-making processes. We emphasize that our methods should be used as analytical assistance tools rather than replacements for human expertise, particularly in high-stakes domains such as healthcare or finance.

This work utilized large language models for writing assistance and language polishing. All technical contributions, experimental results, and scientific insights remain solely the work of the authors.

