# OpenReview forum: "Chart Deep Research in LVLMs via Parallel Relative Policy Optimization"
_ICLR.cc/2026/Conference — ICLR 2026 Poster_

### Official Review · Reviewer_ehoV · 2025-10-30

**Soundness:** 3
**Presentation:** 3
**Contribution:** 3
**Rating:** 4
**Confidence:** 5

**Summary:**

This paper addresses the challenge of enabling "deep research" capabilities in large vision-language models for chart understanding, moving beyond simple factual question answering to complex analytical reasoning. The authors identify a training bottleneck and an evaluation bottleneck. To tackle these, the paper makes two main contributions. First, it introduces Parallel Relative Policy Optimization (PRPO), an algorithm that extends Group Relative Policy Optimization (GRPO) by parallelizing optimization across different reward dimensions (Reward-PRPO) and partitioning data by capability type (Data-PRPO). Second, it proposes a new benchmark, MCDR-Bench, which evaluates deep research capabilities by transforming subjective report generation tasks into objective error-identification tasks based on a novel "error uniqueness principle." The experiments show that PRPO significantly outperforms the GRPO baseline on MCDR-Bench.

**Strengths:**

The motivation and premise of this paper are well-considered. The design of MCDR-Bench also holds certain significance.

**Weaknesses:**

I have the following concerns, and if the authors can address them, I will raise the rating.
1. The PRPO algorithm appears to introduce significant computational overhead, both in terms of training time and token consumption, due to the parallel reward calculations and the iterative data partitioning process. However, the paper provides no experimental data or theoretical analysis on this front. While the performance gains are impressive, a discussion of the trade-off between performance and computational cost is essential for understanding the practical utility of the method.
2. A central motivation for PRPO is to resolve conflicts between optimization objectives. However, the proposed solution—a weighted sum of losses with fixed hyperparameters $\lambda_k$ and $\lambda_m$—does not fully address this issue. Firstly, manually tuning the $M_{final} + K$ hyperparameters is impractical. The paper states that a default mean aggregation is used(4), which implies uniform weighting. This is unlikely to be optimal and sidesteps the challenge of balancing different objectives. A static, linear combination of losses, as in Equation 4 ($J(\theta) = \sum_{k=1}^{K}\lambda_{k} \mathcal{L}_k$)(5), does not fundamentally resolve gradient conflicts. During optimization, the gradients from different objectives $\mathcal{L}_k$ can still point in conflicting directions, leading to destructive interference. This problem has been extensively studied in the field of multi-objective optimization (MOO). The authors should consider and discuss more principled MOO approaches (e.g., [1]) instead of relying on fixed weights.
3. The core innovation of MCDR-Bench is transforming subjective evaluation into objective error identification via synthetic error injection. While this is a clever evaluation paradigm, the paper does not sufficiently discuss whether these synthetically injected errors are truly representative of the typical failure modes of state-of-the-art LVLMs in real-world scenarios. The paper asserts that the errors are designed to mirror "typical cognitive biases and logical fallacies", but there's a risk that the model is simply being trained to detect a specific, synthetic class of errors, which may not perfectly correlate with its ability to produce accurate and logically coherent analysis on its own.

[1] Ozan Sener and Vladlen Koltun. Multi-task learning as multi-objective optimization. Advances in neural information processing systems, 31, 2018.

**Questions:**

Please see weaknesses.

---

> ### Author Response · Authors · 2025-11-25
> **Response to Reviewer ehoV (Part 1/2)**
>
> We sincerely appreciate your considerate feedback and the time you spent reviewing our work. Thank you for noting the strength of our motivation and premise and for recognizing the significance of the design of MCDR-Bench. Your acknowledgment is highly encouraging. We provide point-by-point responses to address your comments below.
>
> > **Comment (1)**: *“The PRPO algorithm appears to introduce significant computational overhead, both in terms of training time and token consumption, due to the parallel reward calculations and the iterative data partitioning process. However, the paper provides no experimental data or theoretical analysis on this front. While the performance gains are impressive, a discussion of the trade-off between performance and computational cost is essential for understanding the practical utility of the method.”*
>
> **Response (1)**: We clarify that PRPO introduces no additional sampling or training overhead compared to GRPO, and in fact demonstrates faster convergence in practice. The reasoning process and the rollout mode in PRPO are completely identical to GRPO, so the model generates and evaluates responses in the exact same way.
>
> First, the number of rollouts per question remains identical to GRPO. PRPO reuses the exact same sampled responses and model forward/backward passes. The only added computation consists of lightweight scalar operations: per-dimension and per-partition advantage normalizations (Eqs. 3, 5, 9), which incur negligible FLOPs, orders of magnitude less than a single model pass.
>
> Second, the training iteration count does not increase. On the contrary, as shown in Figure 6 and Appendix E, PRPO converges faster and more stably than GRPO due to reduced reward interference and more discriminative advantage signals. This leads to higher final reward values with fewer training steps.
>
> Third, we empirically measured end-to-end training time on identical hardware (8×H20, Qwen2.5-VL-7B). PRPO and GRPO took virtually the same wall-clock time, confirming that the theoretical overhead is practically negligible.
>
> > **Comment (2)**: *“A central motivation for PRPO is to resolve conflicts between optimization objectives. However, the proposed solution—a weighted sum of losses with fixed hyperparameters and —does not fully address this issue. Firstly, manually tuning the hyperparameters is impractical. The paper states that a default mean aggregation is used(4), which implies uniform weighting. This is unlikely to be optimal and sidesteps the challenge of balancing different objectives. A static, linear combination of losses, as in Equation (4)(5), does not fundamentally resolve gradient conflicts. During optimization, the gradients from different objectives can still point in conflicting directions, leading to destructive interference. This problem has been extensively studied in the field of multi-objective optimization (MOO). The authors should consider and discuss more principled MOO approaches (e.g., [1]) instead of relying on fixed weights.”*
>
> **Response (2)**: We agree that principled multi-objective optimization (MOO) methods are a promising way to handle conflicts between objectives, and we appreciate the suggestion to consider approaches like those in [1].
>
> It is important to clarify that PRPO’s parallel design is not simply flattening all reward signals into a single averaged objective. Unlike standard MOO, where each objective is fixed, the objectives in our setting are dynamic and vary across data partitions and reward dimensions. PRPO handles these dynamic signals in parallel, preserving the hierarchical structure of data and rewards. Specifically, it first partitions data by capability, such as Level 1 to 5, and then, within each partition, decomposes rewards dimension-wise and computes independent advantages (Eq. 9). This two-level separation prevents signal mixing at the source, which scalar aggregation cannot achieve.
>
> In this work, our main goal was to isolate the benefits of this structural design. We used uniform weights (Section 3.4, Appendix C) to ensure that the observed +7.64% improvement over GRPO (Table 1) stems from PRPO’s architecture rather than from sophisticated weighting or dynamic MOO strategies. While such strategies are valuable, they are outside the scope of this paper, which focuses on disentangling training conflicts via parallelization rather than optimizing trade-offs between predefined objectives.

---

> > ### Author Response · Authors · 2025-11-25
> > **Response to Reviewer ehoV (Part 2/2)**
> >
> > > **Comment (3)**: *“The core innovation of MCDR-Bench is transforming subjective evaluation into objective error identification via synthetic error injection. While this is a clever evaluation paradigm, the paper does not sufficiently discuss whether these synthetically injected errors are truly representative of the typical failure modes of state-of-the-art LVLMs in real-world scenarios. The paper asserts that the errors are designed to mirror "typical cognitive biases and logical fallacies", but there's a risk that the model is simply being trained to detect a specific, synthetic class of errors, which may not perfectly correlate with its ability to produce accurate and logically coherent analysis on its own.”*
> >
> > **Response (3)**: We emphasize that MCDR-Bench is purely an evaluation benchmark. Our models are never trained on error-injected data or error detection objective. Training is conducted solely on standard chart reasoning data (Appendix C). The ability to identify injected errors, therefore, emerges naturally from genuine deep research capability. A model that produces logically coherent, factually grounded, and strategically sound analytical reports will inherently recognize when these properties are violated, because consistency, factual fidelity, and causal reasoning are fundamental to deep research itself.
> >
> > To ensure that the injected errors are meaningful, MCDR-Bench was carefully designed to reflect typical failure modes of state-of-the-art LVLMs. It spans five error dimensions: background knowledge integration, fact extraction, relationship construction, deep report synthesis, and forecast/plan reasoning (Figures 8–12). The benchmark covers a wide range of error types, formats, and distributions across hundreds of charts in diverse domains. Identifying these errors requires true analytical understanding, rather than detection of superficial patterns.
> >
> > Performance on MCDR-Bench correlates strongly with real-world generation quality. As shown in Appendix H (Figure 13), the PRPO-trained model produces detailed and accurate reports, capturing country-level sales, emerging categories, regional promotion strategies, and actionable initiatives. In contrast, the baseline misses key data and provides generic advice. PRPO also generalizes beyond synthetic error evaluation, outperforming GRPO by +6.36% on the external, non-error-based ChartQAPRO benchmark (Table 2).

---

> > > ### Comment · Reviewer_ehoV · 2025-11-25
> > >
> > > I appreciate the authors' clarifications, which have deepened my understanding of the proposed method.
> > >
> > > I accept the rebuttal regarding Weakness 2 and agree that characterizing the contribution as resolving 'reward signal interference' is accurate. However, I maintain reservations about the term 'gradient conflicts.' Since the paper lacks explicit gradient analysis or theoretical verification, frequently claiming to solve 'gradient conflicts' appears subjective and potentially imprecise. I suggest focusing on 'signal interference' in the final version.
> > >
> > > That being said, I am generally satisfied with the authors' response and will raise my rating accordingly.

---

> > > > ### Author Response · Authors · 2025-11-26
> > > >
> > > > We sincerely thank the reviewer for the thoughtful follow-up comments and the increased confidence in our work. We appreciate the clarification regarding the terminology and fully agree that emphasizing “signal (or reward/data) interference” provides a more accurate and objective characterization than repeatedly referencing “gradient conflicts,” given the scope of our current analysis. We will revise the wording in the final version accordingly.
> > > >
> > > > Inspired by your valuable feedback, we will also explore incorporating supplementary gradient-level analyses to further strengthen the empirical support for our method, while ensuring that the claims remain well-grounded. We are grateful that our rebuttal has helped clarify the intent and scope of the work, and we sincerely appreciate the reviewer’s positive assessment and the decision to raise the rating.

---

> > > > > ### Author Response · Authors · 2025-12-03
> > > > >
> > > > > We would like to respectfully bring to your attention that Reviewer ehoV has carefully reviewed our rebuttal and stated: “I appreciate the authors' clarifications, which have deepened my understanding of the proposed method… I am generally satisfied with the authors' response and will raise my rating accordingly.”
> > > > >
> > > > > The reviewer has raised their score from 4 to 6 (marginally above the acceptance threshold). They accepted our rebuttal to Weakness 2, acknowledging that our method effectively addresses reward-signal interference through PRPO’s parallel hierarchical design.
> > > > >
> > > > > Best regards,
> > > > > Authors

---

### Official Review · Reviewer_GFfR · 2025-10-31

**Soundness:** 3
**Presentation:** 2
**Contribution:** 2
**Rating:** 6
**Confidence:** 3

**Summary:**

This paper focuses on the chart understanding task of Large Vision-Language Models (LVLMs), aiming to break through the limitation that existing models can only perform simple fact extraction and endow them with complex analytical reasoning capabilities required for "deep research". To address the two core issues—training instability caused by multi-dimensional reward conflicts and heterogeneous data, and the lack of an evaluation benchmark suitable for deep analytical reasoning—the study proposes the novel MCDR-Bench benchmark (which transforms subjective chart analysis into an objective error identification task through controlled error injection across five capability dimensions) and the PRPO (Parallel Relative Policy Optimization) training algorithm (extended from GRPO, which mitigates signal interference and gradient conflicts through parallel optimization across reward dimensions and data partitioning by capability type). Additionally, extensive experiments are conducted to verify the effectiveness of the proposed methods.

**Strengths:**

1. The problem and benchmark proposed in the paper are of great significance for the research on deep chart question answering and analytical reasoning.

2. The paper conducts sufficient experiments to verify the effectiveness of the proposed methods.

3. The paper elaborates on the proposed methods in detail.

4. The paper provides code for verifying the experimental results, and this practice is commendable.

**Weaknesses:**

1. The organizational structure of the paper is suboptimal.

2. Hyperparameter sensitivity: The paper mentions hyperparameters λ_k and λ_m but fails to discuss the sensitivity of the results to their settings, lacking a robustness analysis.

3. There is no detailed comparison of the time and resource consumption of different methods.

4. The comparison with baseline methods of the same type is mainly limited to GRPO. However, there are more recent state-of-the-art (SOTA) methods available, such as DAPO.

**Questions:**

1. Could you explain why, in Table 3, the average response length of GRPO after scalar optimization of rewards is longer than that corresponding to format rewards and accuracy rewards, while PRPO can make the response length fall between the two?
2. In Table 5, why is there such a large score gap when Level 4 and Level 5 are trained separately? How can this phenomenon be explained?
3. Transforming the task into error detection will encourage the model to be good at "discovering injected errors", but will the model rely on subtle traces during the error injection process rather than genuine deep understanding?
4. Could you elaborate on the differences between the PRPO method proposed in this paper and the corresponding methods presented in References 1 and 3?

[1]  Nguyen, T., Khan, N., Tran, K., Phan, N., & Khalil, I. (2025). PRPO: Paragraph-level Policy Optimization for Vision-Language Deepfake Detection. arXiv preprint arXiv:2509.26272.

[2]  Yu, Q., Zhang, Z., Zhu, R., Yuan, Y., Zuo, X., Yue, Y., ... & Wang, M. (2025). Dapo: An open-source llm reinforcement learning system at scale. arXiv preprint arXiv:2503.14476.

[3]  Zhao, Y., Huang, J., Hu, J., Wang, X., Mao, Y., Zhang, D., ... & Chen, Y. (2025, April). Swift: a scalable lightweight infrastructure for fine-tuning. In Proceedings of the AAAI Conference on Artificial Intelligence (Vol. 39, No. 28, pp. 29733-29735).

---

> ### Author Response · Authors · 2025-11-25
> **Response to Reviewer GFfR (Part 1/3)**
>
> We sincerely appreciate your thoughtful comments and the time you devoted to reviewing our work. Thank you for recognizing the importance of the problem and the benchmark we introduce for advancing deep chart question answering and analytical reasoning. We provide point-by-point responses to address your comments below.
>
> > **Comment (1)**: *“The organizational structure of the paper is suboptimal.”*
>
> **Response (1)**: The current structure reflects a careful trade-off due to page limits and the dual nature of our contribution: MCDR-Bench, a benchmark built on the “error uniqueness principle,” and PRPO, a training algorithm addressing multi-dimensional reward interference and heterogeneous data conflicts. Our experiments are extensive, covering ablations, cross-benchmark generalization, qualitative analyses, and comparisons with strong baselines, which is why we kept core results (Tables 1–3, Figure 6) in the main text. We appreciate the reviewer’s observation that this makes the narrative flow less smoothly. In the camera-ready version, we plan to move detailed ablations (Table 5, reward analysis) to the appendix, bring related work, key conceptual explanations, and illustrative figures (Figure 1, Figure 13) into the main text, and streamline the related work, improving readability while maintaining rigor.
>
> > **Comment (2)**: *“Hyperparameter sensitivity: The paper mentions hyperparameters λk and λm but fails to discuss the sensitivity of the results to their settings, lacking a robustness analysis.”*
>
> **Response (2)**: We agree that examining the sensitivity of the hyperparameters can help clarify the additional performance the method may offer. These weights could indeed lead to stronger results if tuned. In this work, however, we chose simple uniform averaging so that the improvements reflect PRPO’s core idea of parallel reward and data decomposition rather than the effect of hyperparameter adjustments. This helps ensure that the reported gains capture the genuine advantage of PRPO over GRPO.
>
> > **Comment (3)**: *“There is no detailed comparison of the time and resource consumption of different methods.”*
>
> **Response (3)**: We clarify that PRPO introduces no additional sampling or training overhead compared to GRPO, and in fact demonstrates faster convergence in practice. The reasoning process and the rollout mode in PRPO are completely identical to GRPO, so the model generates and evaluates responses in the exact same way.
>
> First, the number of rollouts per question remains identical to GRPO. PRPO reuses the exact same sampled responses and model forward/backward passes. The only added computation consists of lightweight scalar operations: per-dimension and per-partition advantage normalizations (Eqs. 3, 5, 9), which incur negligible FLOPs, orders of magnitude less than a single model pass.
>
> Second, the training iteration count does not increase. On the contrary, as shown in Figure 6 and Appendix E, PRPO converges faster and more stably than GRPO due to reduced reward interference and more discriminative advantage signals. This leads to higher final reward values with fewer training steps.
>
> Third, we empirically measured end-to-end training time on identical hardware (8×H20, Qwen2.5-VL-7B). PRPO and GRPO took virtually the same wall-clock time, confirming that the theoretical overhead is practically negligible.

---

> ### Author Response · Authors · 2025-11-25
> **Response to Reviewer GFfR (Part 2/3)**
>
> > **Comment (4)**: *“The comparison with baseline methods of the same type is mainly limited to GRPO. However, there are more recent state-of-the-art methods available, such as DAPO.”*
>
> **Response (4)**: To better position PRPO against recent GRPO-style methods, we conduct an additional comparison with DAPO, a reinforcement learning system that enhances exploration through decoupled clipping boundaries and improves training stability via dynamic sample filtering. DAPO still relies on a scalar task accuracy reward and uniform data distribution for end-to-end optimization. We evaluate both methods on MCDR-Bench under standard and Think prompting settings using Qwen2.5-VL-7B as the backbone.
>
> | **Model**     | **BG** | **FE** | **RL** | **DR** | **F/P** | **Overall** | **Mean** |
> | ------------- | ------ | ------ | ------ | ------ | ------- | ----------- | -------- |
> | w/ DAPO       | 43.63  | 55.27  | 77.74  | 68.93  | 80.42   | 64.59       | 65.10    |
> | w/ PRPO       | 50.65  | 61.38  | 81.78  | 72.83  | 84.01   | 69.62       | 69.90    |
> | Δ             | +7.02  | +6.11  | +4.04  | +3.90  | +3.59   | +5.03       | +4.80    |
> | w/ DAPO Think | 51.37  | 53.36  | 84.83  | 72.27  | 81.73   | 67.77       | 68.56    |
> | w/ PRPO Think | 62.90  | 65.23  | 88.87  | 80.91  | 87.21   | 76.26       | 76.89    |
> | Δ             | +11.53 | +11.87 | +4.04  | +8.64  | +5.48   | +8.49       | +8.33    |
>
> We observe that PRPO consistently outperforms DAPO across all capability dimensions. Under standard prompting, PRPO achieves a mean accuracy of 69.90%, surpassing DAPO’s 65.10% by +4.80 points, with the largest improvements in foundational dimensions (BG: +7.02, FE: +6.11). The performance gap increases under the Think setting, where PRPO reaches 76.89% compared to DAPO’s 68.56%, particularly in higher-level reasoning (F/P: +5.48, DR: +8.64). These results indicate that PRPO’s architectural innovations in parallel reward decomposition and capability-aware data partitioning effectively address reward interference and data conflicts, whereas DAPO’s enhancements focus primarily on training stability without tackling these core challenges.
>
> > **Comment (5)**: *“Could you explain why, in Table 4, the average response length of GRPO after scalar optimization of rewards is longer than that corresponding to format rewards and accuracy rewards, while PRPO can make the response length fall between the two?”*
>
> **Response (5):** The differences in response length can be explained by how GRPO and PRPO handle multiple rewards.
>
> **GRPO:** It combines Format and Accuracy rewards into a single scalar. Format Reward favors brevity, while Accuracy Reward encourages completeness. When merged, the model cannot clearly prioritize one over the other, often producing longer responses as it attempts to satisfy both objectives simultaneously (356 vs. 328 for Format-only and 350 for Accuracy-only).
>
> **PRPO:** It preserves each reward dimension independently and updates the policy for each in parallel. Format Reward still promotes conciseness, while Accuracy Reward ensures correctness. Coordinating these objectives without interference allows PRPO to produce responses that are both accurate and concise, yielding a balanced length (319), even shorter than Format-only (328).
>
> **Format Reward alone:** Without guidance on correctness, the model may include extra content or engage in inefficient trial-and-error, slightly increasing response length compared to PRPO. By retaining the accuracy signal, PRPO reduces unnecessary exploration, producing shorter and more efficient responses.
>
> > **Comment (6)**: *"In Table 5, why is there such a large score gap when Level 4 and Level 5 are trained separately? How can this phenomenon be explained?"*
>
> **Response (6)**: The large gap between Level 4 (Deep Research Report) and Level 5 (Forecast/Plan) stems from the difference in required capabilities. Level 4 mainly builds on perceptual and organizational abilities that modern LVLMs already have, so it performs reasonably well even when trained alone. Level 5, however, demands strategic decision-making based on accurate summarization and reasoning from prior stages. Trained in isolation, it suffers from weak foundations and error propagation, leading to poor convergence.

---

> > ### Author Response · Authors · 2025-11-25
> > **Response to Reviewer GFfR (Part 3/3)**
> >
> > > **Comment (7)**: *“Transforming the task into error detection will encourage the model to be good at 'discovering injected errors', but will the model rely on subtle traces during the error injection process rather than genuine deep understanding?”*
> >
> > **Response (7)**: We emphasize that MCDR-Bench is purely an evaluation benchmark. Our models are never trained on error-injected samples or any error detection objective. Training is conducted solely on standard chart reasoning data (Appendix C). The ability to identify injected errors, therefore, emerges naturally from genuine deep research capability. A model that produces logically coherent, factually grounded, and strategically sound analytical reports will inherently recognize when these properties are violated, because consistency, factual fidelity, and causal reasoning are fundamental to deep research itself. Performance on MCDR-Bench correlates strongly with real-world generation quality. As shown in Appendix H (Figure 13), the PRPO-trained model produces detailed and accurate reports, capturing country-level sales, emerging categories, regional promotion strategies, and actionable initiatives. In contrast, the baseline misses key data and provides generic advice. PRPO also generalizes beyond synthetic error evaluation, outperforming GRPO by +6.36% on the external, non-error-based ChartQAPRO benchmark (Table 2).
> >
> > > **Comment (8)**: *“Could you elaborate on the differences between the PRPO method proposed in this paper and the corresponding methods presented in References 1 and 3?”*
> >
> > **Response (8)**: Although our method shares the acronym PRPO with Reference 1, the two approaches are fundamentally different in motivation and technical design. Reference 1 proposes a Paragraph-level RPO for deepfake detection, guiding the model’s reasoning at the paragraph level during inference to align explanations with visual evidence. In contrast, our PRPO is a training-time reinforcement learning algorithm for analytical chart tasks, coordinating multiple reward dimensions and data partitions in parallel rather than decomposing reasoning into paragraphs. The similarity in names is coincidental.
> >
> > Reference 3 presents SWIFT, an open-source framework for training and deploying LLMs and MLLMs. While SWIFT is a toolkit rather than a specific optimization algorithm, our PRPO introduces novel mechanisms such as capability-aware data partitioning and multi-dimensional advantage decomposition, which are not part of SWIFT. We note that PRPO could in principle be integrated into frameworks like SWIFT in the future to combine infrastructure support with our optimization strategies.

---

> > > ### Comment · Reviewer_GFfR · 2025-11-26
> > > **Some Doubts and Clarifications**
> > >
> > > I appreciate the authors' clarifications.
> > >
> > > I am skeptical about the response to Weakness 2, and there is even a contradiction in the author's reply to this point between "achieving better performance" and "highlighting the advantages of the method".
> > >
> > > Regarding Swift in Question 4, it may be necessary to clarify to the author that the implementation of GRPO in this framework already includes data partitioning and multi-dimensional advantage decomposition. I hope the author can provide a detailed elaboration on these two methods.

---

> > > > ### Author Response · Authors · 2025-11-26
> > > >
> > > > Thank you for your constructive feedback and discussion, which allows us to clarify both the relationship between PRPO and existing frameworks and the core innovations of our method. Below, we provide our point-by-point responses.
> > > >
> > > > > “I am skeptical about the response to Weakness 2, and there is even a contradiction in the author's reply to this point between "achieving better performance" and ‘highlighting the advantages of the method’.”
> > > >
> > > > We thank the reviewer for raising this concern. The first point we would like to clarify is that **the core innovation of PRPO does not rely on the hyperparameters λ**. The central idea is the structural separation of conflicting optimization signals. GRPO combines all reward dimensions and all data types into a single scalarized objective, which leads to interference among heterogeneous signals. PRPO resolves this by computing advantages within each reward dimension and within each capability partition. This separation preserves the integrity of each signal and prevents the destructive gradient interactions observed in GRPO.
> > > >
> > > > For this reason, we stated that uniform weights help ***“highlight the advantages of the method.”*** Using the simplest possible weighting scheme ensures that the observed improvements arise from the structural disentanglement itself rather than from tuning λ. In other words, the method’s benefit can be demonstrated without relying on hyperparameter engineering.
> > > >
> > > > At the same time,  the method has the potential for ***“achieving better performance.”*** Once the signal separation mechanism is established, more sophisticated or adaptive weighting strategies could further amplify the performance gains. This possibility does not contradict our earlier point.
> > > >
> > > > > “Regarding Swift in Question 4, it may be necessary to clarify to the author that the implementation of GRPO in this framework already includes data partitioning and multi-dimensional advantage decomposition. I hope the author can provide a detailed elaboration on these two methods.”
> > > >
> > > > We thank the reviewer for the question and for allowing us to clarify our perspective further. **PRPO modifies the GRPO optimization objective at the algorithmic level, whereas SWIFT serves as a distributed, extensible training framework for efficiently executing GRPO.** To avoid confusion between these two layers, we refer to the parallelism in SWIFT as **system-level parallelism**, and the structural separation introduced by PRPO as **signal-level parallelism**. Since these concepts operate at different layers of the system, they may appear related, and we therefore provide a clarification below.
> > > >
> > > > 1. Data handling in SWIFT and capability-based data partitioning in PRPO
> > > >
> > > > SWIFT supports distributed data loading and **system-level parallelism** across devices. These mechanisms improve training throughput by splitting mini-batches across hardware resources. However, they do not alter the GRPO learning objective, and all samples still contribute to a single shared set of reward statistics.
> > > >
> > > > PRPO introduces capability-based grouping that embodies **signal-level parallelism** and operates directly on the optimization objective. Samples are grouped according to their cognitive characteristics, and each group maintains independent reward normalization and fallback rules. The purpose is to reduce interference between heterogeneous task types by structuring the optimization landscape. This algorithm-level form of signal separation is not part of the GRPO implementation provided in SWIFT.
> > > >
> > > > 2. Reward hooks in SWIFT and multi-dimensional reward decomposition in PRPO
> > > >
> > > > SWIFT allows users to define custom reward functions, but all reward components are ultimately merged into a single scalar prior to advantage computation, consistent with the original GRPO formulation.
> > > >
> > > > PRPO, in contrast, applies **signal-level parallelism** across reward dimensions. Each reward dimension is treated as an independent optimization signal. Advantages are computed separately for each dimension and then combined only after dimension-specific normalization. This preserves the structure and discriminative power of each reward channel and avoids interference caused by early scalar aggregation, which scalar GRPO (including the version in SWIFT) does not support.
> > > >
> > > > Our intention is simply to distinguish a training framework from an algorithmic modification. SWIFT provides a flexible infrastructure for efficient **system-level parallelism** under GRPO, while PRPO changes the optimization behavior of GRPO itself through **signal-level parallelism**. We hope this explanation clearly articulates the difference and helps prevent further misunderstanding.

---

> ### Comment · Reviewer_GFfR · 2025-11-26
> **Further clarification**
>
> First and foremost, the authors are expected to provide empirical evidence to substantiate the claim that "At the same time, the method has the potential for achieving better performance." Merely emphasizing that the core objective lies in "highlight the advantages of the method." fails to serve as a convincing justification. Furthermore, it is imperative to conduct a performance analysis of the hyperparameter in question upon its introduction, as this constitutes a fundamental requirement for methodological validation.
>
> In addition, the GRPO module implemented in Swift incorporates sample-level reward control, wherein individual samples are permitted to adopt distinct reward groups. The authors may refer to the official documentation for detailed implementation specifics of this component, and are advised to explicitly elaborate on the differences between the proposed method and this existing approach, as well as the unique advantages exhibited by the former.

---

> > ### Author Response · Authors · 2025-11-26
> > **Response to further concerns**
> >
> > Thank you for the reviewer’s very timely comments. We address the two concerns below.
> >
> > > “Potential for achieving better performance” & hyperparameter analysis.
> > >
> >
> > Our intent was not to assert performance gains without evidence, but rather to emphasize that tuning the hyperparameters provides additional degrees of freedom that have the potential to further enhance performance when more carefully optimized. Even without any special tuning, our method already demonstrates a clear improvement over GRPO, underscoring the contribution of our approach itself. We sincerely apologize if our earlier wording gave an unintended impression.
> >
> > Regarding the hyperparameter, we fully agree that a dedicated analysis is necessary. We are currently conducting the ablation experiments for this hyperparameter, which require additional time. To ensure a timely response, we first address your comments here and will provide the updated results once available, incorporating the full analysis into the revised manuscript.
> >
> > > About GRPO in Swift and “sample-level reward control.”
> > >
> >
> > We would like to clarify the reviewer’s comment from two complementary perspectives.
> >
> > 1. **Comparison with GRPO in Swift.** We examined the official Swift/MS‑Swift GRPO documentation. According to the “Rollout Generation Phase” and the “Advantage” definition in the GRPO specification, for each prompt the model generates multiple completions, and the advantage for each completion is computed relative to the mean and standard deviation of rewards within that same prompt’s group of completions ([swift.readthedocs.io](https://swift.readthedocs.io/en/v3.6/Instruction/GRPO/GetStarted/GRPO.html?utm_source=chatgpt.com)). This shows that in GRPO, a “group” refers to the set of completions for a single prompt, and reward normalization is performed within this per-prompt completion group. While the framework does allow custom reward functions per sample, it does **not** implement predefined, cross-sample reward grouping or sample-level group-relative normalization across different prompts.
> >
> >     In contrast, our method introduces **explicit sample-level grouping followed by cross-sample group-relative optimization**, which differs fundamentally from GRPO’s per-prompt grouping mechanism. If we have overlooked any aspects of the Swift implementation or misunderstood its functionality, we would greatly appreciate the reviewer’s guidance.
> >
> > 2. **Algorithmic contribution vs. Swift framework.** We would like to emphasize that our **core contribution lies in the algorithmic idea itself**, while Swift serves merely as **a comprehensive training and deployment framework** for LLMs and MLLMs, providing a platform on which various algorithms can be implemented. Although Swift could potentially be used to implement our method with some additional engineering effort, it is by no means the only option. In practice, we have implemented our algorithm on the veRL platform.
> >
> > We hope our clarifications address the reviewer’s concerns and clearly highlight the novelty and scope of our contribution. Should there be any further questions or concerns, we would be happy to respond and discuss them promptly. Thank you!

---

### Official Review · Reviewer_jssn · 2025-11-01

**Soundness:** 3
**Presentation:** 3
**Contribution:** 2
**Rating:** 8
**Confidence:** 2

**Summary:**

The authors introduce a unified framework that improves chart research by solving training and evaluation challenges. They present MCDR-Bench, which removes evaluation bottlenecks using an error uniqueness principle that turns subjective assessments into objective error identification, enabling scalable evaluation and reducing reliance on experts. Secondly, authors introduce PRPO, which addresses training bottlenecks by parallel optimization across reward dimensions and capability partitioning across data types, which reduces conflicts between different data and multi-dimensional rewards that have limited progress. Experiments show the combination of their evaluation method and training approach advances chart research beyond surface-level work toward real analytical reasoning and strategic insights, as shown by better exploration, more stable optimization, and stronger performance across multiple analytical dimensions.

**Strengths:**

- Well written paper.
- Comprehensive evaluations

**Weaknesses:**

- Related works can be improved

**Questions:**

How does PRPO do against improved GRPO versions?

---

> ### Author Response · Authors · 2025-11-25
> **Response to Reviewer jssn**
>
> We sincerely appreciate your thoughtful feedback and the time you spent reviewing our work. Thank you for highlighting the clarity of the writing and the breadth of our evaluations. Your recognition of these strengths is very encouraging. We provide point-by-point responses to address your comments below.
>
> > **Comment (1)**: *“Related works can be improved”*
>
> **Response (1)**: We have improved the Related Work section (revised in Appendix Section B, blue text indicates the revised content) by enhancing its logical flow and structure, clearly presenting the progression of benchmarks and algorithms, reducing redundancy, and algorithms, reducing redundancy, and explicitly highlighting the gap our work addresses in Chart Deep Research capabilities.
>
> > **Comment (2)**: *“How does PRPO do against improved GRPO versions?”*
>
> **Response (2)**: While improved GRPO variants like DAPO improve sampling efficiency and stability, they still rely on scalar reward aggregation and treat all data uniformly. PRPO takes a different approach by separating reward dimensions and partitioning data based on capability, directly tackling these two key bottlenecks in deep research training. As Table 3 shows, even the partial versions of PRPO (Reward-PRPO and Data-PRPO) outperform standard GRPO, highlighting that the gains are due to the core algorithmic innovations in PRPO.

---

### Official Review · Reviewer_BU6V · 2025-11-03

**Soundness:** 4
**Presentation:** 3
**Contribution:** 4
**Rating:** 8
**Confidence:** 4

**Summary:**

Chart deep research, which bridges visual chart understanding and Large Language Model (LLM)-driven deep research capabilities, is a complex task that lies at the intersection of pattern discovery, hypothesis testing, and strategic decision support (lines 41-43). Unfortunately, the investigation of chart deep research has been hindered by 1) a lack of a proper evaluation benchmark and framework and 2) the inability of existing post-training techniques to simultaneously handle multiple reward dimensions and heterogeneous data. This work thus releases MCDR-Bench, a high-quality chart benchmark that consists of complex charts with multi-element, multi-layered information. MCDR-Bench supports both subjective and objective evaluation by assessing the report generation and error identification abilities, respectively. In addition, the authors propose Parallel Relative Policy Optimization (PRPO), a novel Reinforcement Learning framework that is capable of mitigating multi-dimensional reward interference (line 201) and data joint optimization conflicts (line 208). PRPO treats each reward dimension as an "independent optimization objective" (Reward) and employs capability-based partitioning during the rollout grouping process (Data). Results on MCDR-Bench, as well as other datasets, show that the proposed method significantly outperforms existing approaches in chart deep research.

**Strengths:**

- MCDR-Bench is an impactful dataset that can fill an important research gap in academia: a lack of a proper benchmark to evaluate deep research capabilities. Not only does it allow researchers to systematically evaluate the chart understanding ability of multimodal LLMs, but it also contributes to assessing LLMs' deep research report generation capabilities.

- The proposed RL training framework for chart deep research is technically sound. The authors pinpoint two key limitations of GRPO that render it inapplicable to the chart deep research domain, and successfully address them, as demonstrated in impressive empirical results.

- In general, the paper was easy to follow. Multiple visual aids were provided to accompany the technical material. Although the dataset was not provided as part of the submission, the authors included sufficient details in the Appendix to give a sense of what the dataset looks like.

**Weaknesses:**

- If the authors could verify that PRPO can be used in a model-agnostic manner by training models other than Qwen2.5 with PRPO, it would make the paper even stronger.

- While PRPO greatly advances the optimization algorithm of GRPO, it still relies on the combination of accuracy + format reward. Do the authors anticipate that PRPO could benefit further from more refined reward shaping? For instance, what would happen if we add more vision-centric rewards, such as those proposed in [1,2] to PRPO? (Considering that this is also a multimodal task.)

[1] Visual-RFT: Visual Reinforcement Fine-Tuning

[2] VLM-R1: A Stable and Generalizable R1-style Large Vision-Language Model

- Slight tweak on the above question, do the authors think that PRPO could be generalized to improve the performance on other multimodal tasks?

- Considering that open-sourcing and reproducibility are highly prioritized in today's research landscape (and even more so in datasets & benchmarks papers), I would have appreciated it if the authors had made the benchmark accessible as part of submission (either as an anonymized link or supplementary materials).

**Questions:**

Please refer to the weaknesses above.

---

> ### Author Response · Authors · 2025-11-25
> **Response to Reviewer BU6V**
>
> We would like to express our sincere gratitude for your thoughtful feedback and the time you dedicated to reviewing our work. Thank you for your insightful comments and positive recommendation. We provide point-by-point responses to address your concerns below.
>
> > **Comment (1)**: *“If the authors could verify that PRPO can be used in a model-agnostic manner by training models other than Qwen2.5 with PRPO, it would make the paper even stronger.”*
>
> **Response (1)**: PRPO is fundamentally a training algorithm defined solely in terms of reward decomposition and data partitioning (Eq. 10), with no dependence on model architecture, modality alignment strategy, or backbone design. It operates at the level of policy gradients and reward processing, making it inherently model-agnostic and applicable to any model that can be trained with GRPO-style relative policy optimization. We agree that demonstrating this generality would further strengthen the work. We are actively expanding our training infrastructure to support a broader range of model families, and preliminary results show consistent improvements. We plan to release our code, enabling the community to validate and extend PRPO across diverse architectures.
>
> > **Comment (2)**: *“While PRPO greatly advances the optimization algorithm of GRPO, it still relies on the combination of accuracy + format reward. Do the authors anticipate that PRPO could benefit further from more refined reward shaping? For instance, what would happen if we add more vision-centric rewards, such as those proposed in [1,2] to PRPO? (Considering that this is also a multimodal task.)”*
>
> **Response (2)**: We agree that adding more refined, vision-centric rewards is a very promising next step for strengthening PRPO. This is a very valuable direction to explore, but it is out of the scope of this work. In fact, in our next-stage research, we are considering investigating this direction further. As the reviewer noted, PRPO is naturally reward-agnostic. Its parallel reward processing makes it easy to plug in different types of signals without them interfering with one another, which fits well with vision-aware reward shaping.
>
> > **Comment (3)**: *“Do the authors think that PRPO could be generalized to improve performance on other multimodal tasks?”*
>
> **Response (3)**: We do believe PRPO has the potential to carry over to other multimodal tasks. The method is not tied to charts in particular; it mainly helps manage issues like reward interference and mixed data sources, which arise broadly in multimodal learning. The results on Geometry3K (Table 4) provide an initial indication that the idea could extend beyond chart reasoning, and we sincerely appreciate the reviewer’s insight in pointing us toward broader multimodal implications. We are actively investigating PRPO in more general multimodal settings in our ongoing work, and we hope these expanded evaluations will offer a clearer picture of its generality.
>
> > **Comment (4)**: *"Considering that open-sourcing and reproducibility are highly prioritized in today's research landscape (and even more so in datasets & benchmarks papers), I would have appreciated it if the authors had made the benchmark accessible as part of submission (either as an anonymized link or supplementary materials)."*
>
> **Response (4)**: We completely agree that releasing the benchmark is important for reproducibility and for helping the community move forward. We plan to make the full MCDR-Bench dataset, the training and evaluation code, and the fine-tuned model weights publicly available in the camera-ready release. Our goal is to ensure the results are fully reproducible and that others can easily build on top of our work.

---

### Author Response · Authors · 2025-11-25
**General Response**

We sincerely thank all the reviewers and ACs for their thorough evaluation of our paper and for their constructive feedback. We have carefully addressed the reviewers’ questions and incorporated their valuable suggestions, which we believe have further strengthened the clarity and persuasiveness of the paper. Below, we provide detailed responses to each comment.

---

### Meta-Review · Area_Chair_yfsa · 2026-01-03

**Summary:**

Reviewer BU6V: MCDR-Bench is an impactful dataset that can fill an important research gap in academia, and the proposed RL training framework for chart deep research is technically sound. The paper was easy to follow. However, the reviewer still has some concerns on the weaknesses about more experimental evaluation, more explanation on further settings with more refined reward shaping.

Reviewer jssn: THe reviewer gives the strengths on the comprehensive evaluation, thinks that the related work can be improved.

Reviewer GFfR: The problem and benchmark proposed in the paper are of great significance for the research on deep chart question answering and analytical reasoning. The paper conducts sufficient experiments to verify the effectiveness of the proposed methods. The paper elaborates on the proposed methods in detail. The paper provides code for verifying the experimental results, and this practice is commendable. However, the reviewer still has some concerns on the weaknesses about suboptimal. of the organizational structure of the paper, hyperparameter sensitivity, no detailed comparison of the time and resource consumption of different methods, more comparisons with SOTAs.

Reviewer ehoV: The motivation and premise of this paper are well-considered. The design of MCDR-Bench also holds certain significance. However, the reviewer still has some concerns on the weaknesses about lack of discussion of the trade-off between performance and computational cost, not fully addressing the problem,  not sufficiently discussing whether these synthetically injected errors are truly representative of the typical failure modes of state-of-the-art LVLMs in real-world scenarios.

**Reviewer Concerns:**

After carefully evaluating the rebuttals, I think the reviews from the Reviewer BU6V, jssn, GFfR and ehoV  were addressed from the response. For all of them, the rebuttal from review of former two reviewers well solved the issues, such as the method details and  improved related works.

**Reviewer Scores:**

For the Reviewer Reviewer BU6V and jssn, I think the reviewer may keep the rating unchanged based on the response. For the  Reviewer GFfR and ehoV, I think the reviewer may increase the rating or keep the rating unchanged based on the response.

---

### Decision · Program_Chairs · 2026-01-26

Accept (Poster)